# Defective synaptic transmission causes disease signs in a mouse model of juvenile neuronal ceroid lipofuscinosis

Benedikt Grünewald[1,2,3], Maren D Lange[4], Christian Werner[1,3], Aet O'Leary[5], Andreas Weishaupt[3], Sandy Popp[6], David A Pearce[7], Heinz Wiendl[3,8], Andreas Reif[5], Hans C Pape[4], Klaus V Toyka[3], Claudia Sommer[3], Christian Geis[1,2,3]*

[1]Hans-Berger Department of Neurology, Jena University Hospital, Jena, Germany; [2]Integrated Research and Treatment Center—Center for Sepsis Control and Care, Jena University Hospital, Jena, Germany; [3]Department of Neurology, University Hospital Würzburg, Würzburg, Germany; [4]Institute of Physiology I, University of Münster, Münster, Germany; [5]Department of Psychiatry, Psychosomatic Medicine and Psychotherapy, University Hospital Frankfurt, Frankfurt, Germany; [6]Department of Psychiatry, Psychosomatics and Psychotherapy, University Hospital Würzburg, Würzburg, Germany; [7]Sanford Children's Health Research Center, Sanford Research, Sioux Falls, United States; [8]Department of Neurology, University of Münster, Münster, Germany

*For correspondence:
christian.geis@med.uni-jena.de

Competing interests: The authors declare that no competing interests exist.

**Abstract** Juvenile neuronal ceroid lipofuscinosis (JNCL or Batten disease) caused by mutations in the *CLN3* gene is the most prevalent inherited neurodegenerative disease in childhood resulting in widespread central nervous system dysfunction and premature death. The consequences of *CLN3* mutation on the progression of the disease, on neuronal transmission, and on central nervous network dysfunction are poorly understood. We used *Cln3* knockout (*Cln3^{Δex1-6}*) mice and found increased anxiety-related behavior and impaired aversive learning as well as markedly affected motor function including disordered coordination. Patch-clamp and loose-patch recordings revealed severely affected inhibitory and excitatory synaptic transmission in the amygdala, hippocampus, and cerebellar networks. Changes in presynaptic release properties may result from dysfunction of CLN3 protein. Furthermore, loss of calbindin, neuropeptide Y, parvalbumin, and GAD65-positive interneurons in central networks collectively support the hypothesis that degeneration of GABAergic interneurons may be the cause of supraspinal GABAergic disinhibition.
DOI: https://doi.org/10.7554/eLife.28685.001

## Introduction

Juvenile neuronal ceroid lipofuscinosis (JNCL or Batten disease) is the most prevalent inherited neurodegenerative disease in childhood caused by autosomal recessive loss-of-function mutations in the *CLN3* gene. The clinical syndrome consists of gradual visual loss, motor and cognitive decline, ataxia, seizures, psychiatric abnormalities, and death in the third or fourth decade of life (*Jalanko and Braulke, 2009*; *Adams et al., 2007*; *Haltia and Goebel, 2013*). The CLN3 protein is a transmembrane protein (*Ratajczak et al., 2014*) localized in synapses, in early endosomes (*Uusi-Rauva et al., 2012*), and in the Golgi apparatus (*Getty et al., 2013*). Mutations in the *CLN3* gene and its homologues have been analyzed in various species including mice, *Drosophila,* and yeast, revealing multiple biochemical defects and interactions at the cellular level (*Tuxworth et al., 2009*;

*Luiro et al., 2006*; *Chandrachud et al., 2015*; *Mitchison et al., 1999*). The existing mouse models for JNCL are based on a *Cln3* deletion (*Cln3*$^{\Delta ex1-6}$) or on knock-in mutations of *Cln3* (e.g. *Cln3*$^{\Delta ex7/8}$). They are valuable tools for studying JNCL, since they resemble many phenotypic and histopathological abnormalities of the human disease, and these deficits can be attenuated by AAV-mediated restoration of CLN3 levels (*Eliason et al., 2007*; *Katz et al., 1999*; *Kovács and Pearce, 2015*; *Lim et al., 2007*; *Weimer et al., 2009*; *Bosch et al., 2016*). How these molecular changes induce the neurological symptoms of the JNCL is still unknown. Recent studies using extracellular field potential recordings showed that axonal excitability is altered in the hippocampal CA1-region and the visual cortex of the *Cln3*$^{\Delta ex7/8}$ mouse model (*Burkovetskaya et al., 2017*). It is speculated that early changes in the balance of glutamate and GABA may contribute to seizure development and neurodegeneration, as neurons in *Cln3*$^{\Delta ex1-6}$ mice seem to be exceptionally vulnerable to glutamate excitotoxicity (*Pears et al., 2005*; *Finn et al., 2011*; *Kovács et al., 2006*). Counteracting this imbalance by pharmacological means has an ameliorating effect on disease symptoms (*Kovács et al., 2012*, *2011*). To date, the consequences of these alterations in transmitter activity on neuronal and CNS network function have not yet been elucidated and direct experimental evidence of synaptic dysfunction caused by dysfunctional CLN3 is lacking.

Here, we studied the *Cln3*$^{\Delta ex1-6}$ mouse model of JNCL with a combined approach of neurophysiological, behavioral, and immunopathological investigations analyzing synaptic and network function in the CNS. We found that synaptic transmission is severely disturbed in the amygdala, hippocampus, and cerebellum accounting for progressive disease signs in the *Cln3*$^{\Delta ex1-6}$ mouse model. Moreover, the loss of GABAergic interneurons in the amygdala-hippocampal complex may cause increased anxiety-like behavior and defective aversive learning. Disinhibition of Purkinje cells (PCs) may be responsible for cerebellar ataxia.

## Results

### Synaptic transmission defects in the amygdala are associated with anxiety related behavior in *Cln3*$^{\Delta ex1-6}$ mice

Psychiatric abnormalities and anxiety are common features and a major cause of disability in JNCL patients (*Bäckman et al., 2001, 2005*). We first evaluated anxiety- and exploration-related behavior of Cln3$^{\Delta ex1-6}$ mice and wild-type (wt) littermates. We used the dish test for repetitive non-invasive testing of anxiety related behavior and found a prolonged time until escape of Cln3$^{\Delta ex1-6}$ mice beginning at an age of 7 months, which remained stable at later time points (*Figure 1A*). In the open-field (OF) and elevated plus maze (EPM) mice were tested at the age of 7 and 14 months. We observed markedly reduced center visits and center times of Cln3$^{\Delta ex1-6}$ mice as compared to wt controls in the OF at the age of 14 months while there were no differences in younger mice (*Figure 1B*, *Figure 1—figure supplement 1A*). Accordingly, 14 months but not 7 months old Cln3$^{\Delta ex1-6}$ mice made fewer visits to the open arms and spent more time in the closed arms in the EPM. Body stretching as an indicator of explorative behavior was significantly reduced in the Cln3$^{\Delta ex1-6}$ mice (*Figure 1C*, *Figure 1—figure supplement 1B*).

In principle, anxious behavior may be due to dysfunction of synaptic transmission in limbic structures, for example the amygdala-hippocampus complex. Therefore, we tested synaptic transmission in the lateral amygdala (LA) as the central region for processing of fear-related signals at the age of 14 months. We performed whole-cell recordings from LA principle neurons (PNs) in acute amygdala slice preparations (schematically shown in *Figure 2A*) and analyzed GABAergic inhibitory postsynaptic currents (IPSCs) as well as glutamatergic excitatory postsynaptic currents (EPSCs). Membrane potential and input resistance were unchanged in both genotypes (membrane potential $-67 \pm 0.9$ mV vs. $-67 \pm 0.8$ mV; input resistance $378 \pm 23$ MΩ vs. $330 \pm 23$ MΩ; n = 33/7 vs. 27/6 [number of recordings/number of mice]; Cln3$^{\Delta ex1-6}$ mice vs. wt mice). The IPSCs were recorded in the presence of DNQX and AP5, the EPSCs in the presence of bicucullin. The frequency of miniature (mIPSCs; in the presence of TTX) and spontaneous (sIPSCs) events was significantly reduced in LA PNs from Cln3$^{\Delta ex1-6}$ mice as compared to wt littermates, whereas the mean amplitude was significantly different between genotypes for the sIPSC but not mIPSCs (*Figure 2B and C*). The amplitude of evoked inhibitory postsynaptic responses (eIPSCs) through local microstimulation in LA was significantly reduced in Cln3$^{\Delta ex1-6}$ mice (*Figure 2A and D*, *Figure 2—figure supplement 1*) thus indicating

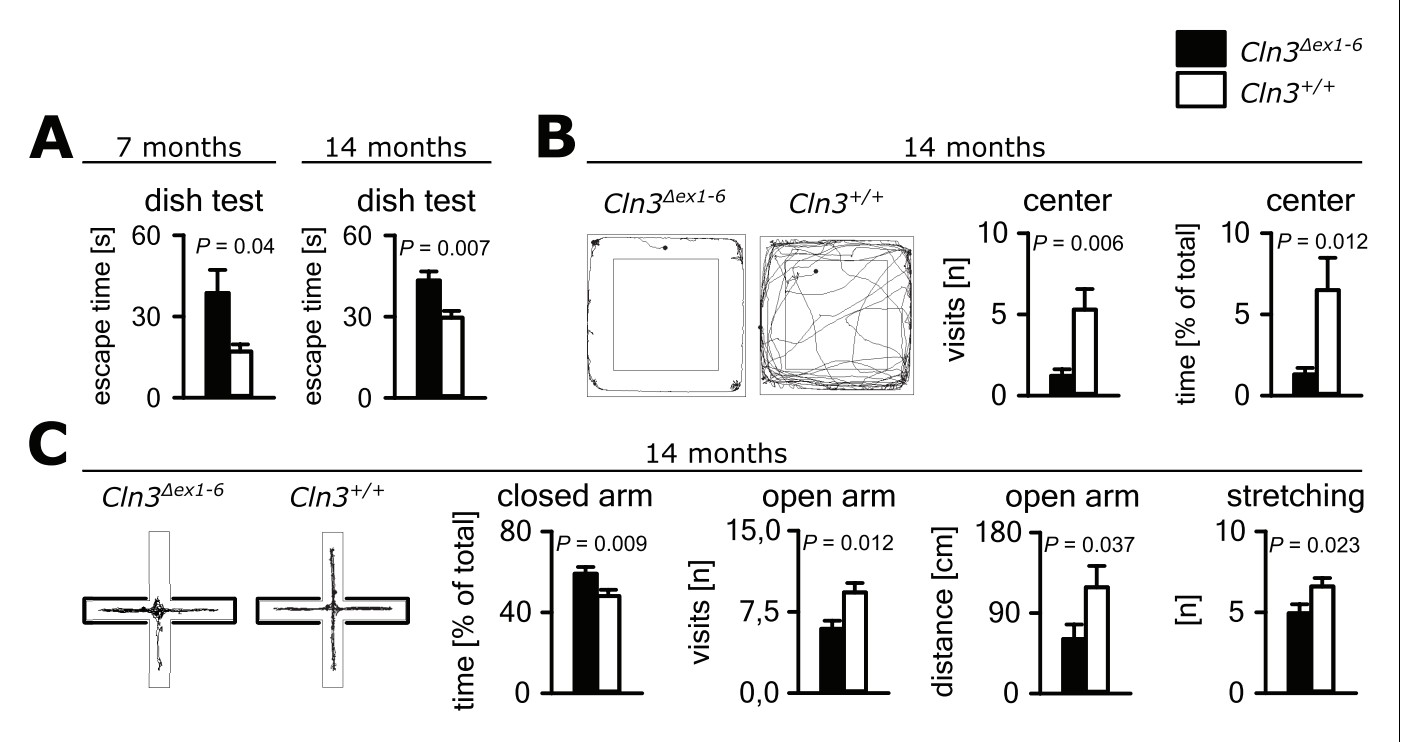

**Figure 1.** Anxiety-related behavior in *Cln3*$^{\Delta ex1-6}$ mice. (**A**) Escape time to cross the rim barrier of a petri dish was increased in *Cln3*$^{\Delta ex1-6}$ mice compared to wt littermates indicating reduced explorative behavior (wt: *n* = 15/24 [7 month/14 month]; Cln3$^{\Delta ex1-6}$: *n* = 14/18; Mann-Whitney Test). (**B**) Increased anxiety-related behavior of *Cln3*$^{\Delta ex1-6}$ mice at the age of 14 months in the open field (OF) test. Representative tracks and data plots for OF related to genotypes. Note that *Cln3*$^{\Delta ex1-6}$ mice visited the center area in the OF less frequently and spent more time in the peripheral areas (n = 18 vs. 20; Mann-Whitney test). (**C**) Increased anxiety-related behavior in the elevated plus maze (EPM) test at the age of 14 months. In the EPM *Cln3*$^{\Delta ex1-6}$ mice showed less visits and reduced travel distance in the open arms and preferred to stay in the closed arms indicating increased anxiety-related behavior. Microbehavior analysis revealed reduced stretchings confirming reduced explorative drive in both test paradigms (n = 18 vs. 20; open arm distance: Mann-Whitney, all other Student's t-test). Exact values of N, dispersion and precision measures are provided in ***Supplementary file 2***.

DOI: https://doi.org/10.7554/eLife.28685.002

The following figure supplement is available for figure 1:

**Figure supplement 1.** Open field (OF) and elevated plus maze testing at the age of 7 months.

DOI: https://doi.org/10.7554/eLife.28685.003

reduction of GABAergic afferent input to LA PNs. The frequency of miniature and spontaneous excitatory postsynaptic currents (mEPSCs and sEPSCs) was reduced in LA - PNs of *Cln3*$^{\Delta ex1-6}$ mice while amplitudes were unchanged (***Figure 2E and F***). Similar to the findings for GABAergic transmission, the peak amplitude of the evoked (e) EPSCs was reduced in *Cln3*$^{\Delta ex1-6}$ mice, pointing to a reduction of excitatory inputs to LA PNs (***Figure 2A and G***, ***Figure 2—figure supplement 2***).

GABAergic disinhibition in the LA is associated with increased anxiety-related behavior. Several types of interneurons contribute to information processing in the amygdala and project onto LA PNs (***Ehrlich et al., 2009***). Quantitative analysis of different interneuron subpopulations as identified by their immunohistochemical markers revealed a reduced number of calbindin (Calb$^+$), parvalbumin (Parv$^+$) and neuropeptide Y (NPY$^+$)- positive interneurons within the amygdala of 14-month-old *Cln3*$^{\Delta ex1-6}$ mice compared to wt littermates (***Figure 3***). The overall number of neurons was unchanged in both genotypes as identified by their NeuN immunoreactivity (***Figure 3—figure supplement 1***).

### Hippocampal synaptic transmission is impaired in *Cln3*$^{\Delta ex1-6}$ mice and cued and contextual fear memory is reduced

Cognitive decline is one of the hallmarks of JNCL patients and equivalent memory impairment was also observed in a *Cln3*$^{\Delta ex1-6}$ mouse model (***Katz et al., 1999***; ***Wendt et al., 2005***). To assess

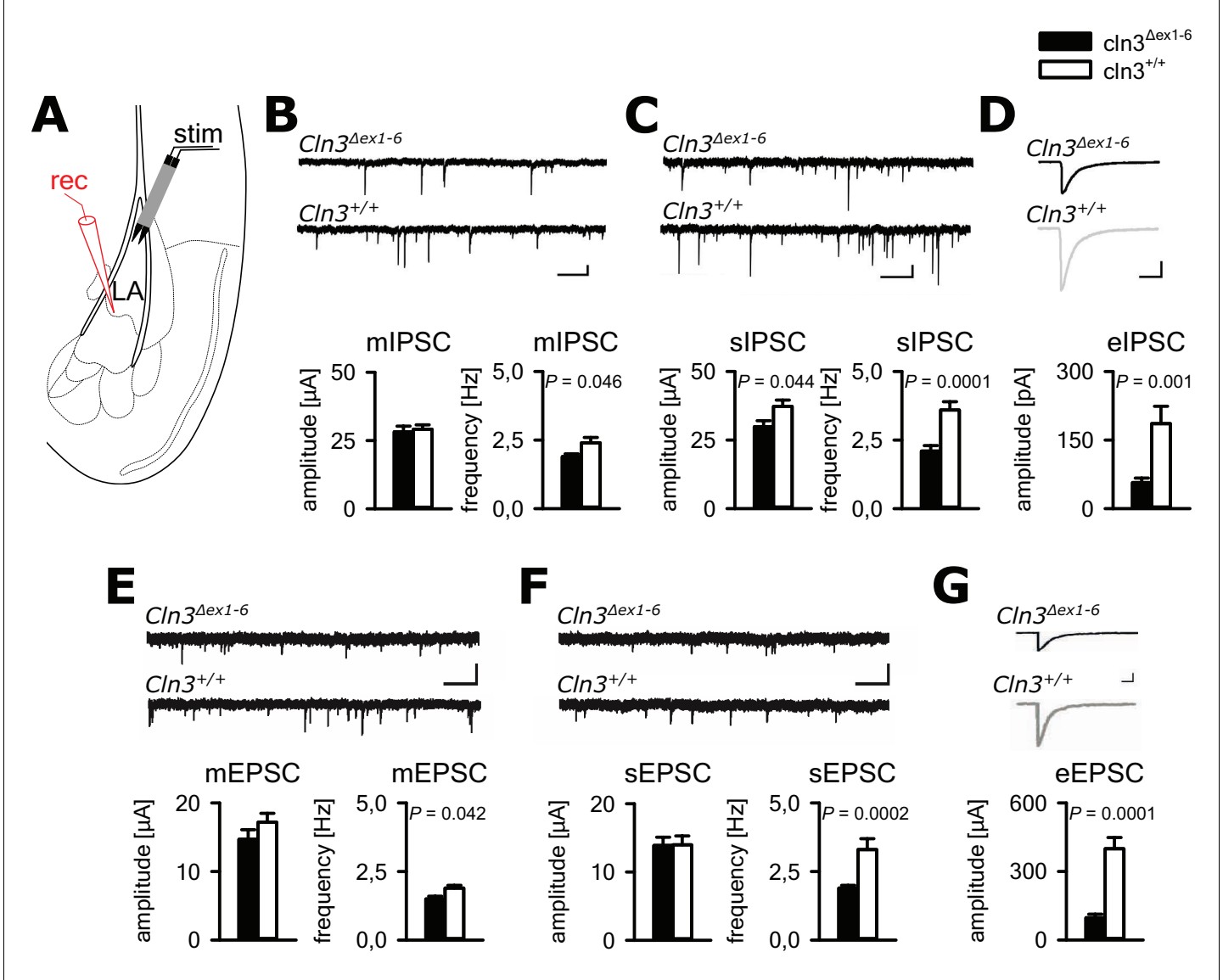

**Figure 2.** Defective synaptic transmission in the amygdala of 14 months old $Cln3^{\Delta ex1-6}$ mice. (**A**) Schematic illustration of the recording situation in the lateral amygdala (LA). Whole-cell patch recordings were performed from principal neurons (PN) in the LA (rec) by local stimulation of GABAergic or glutamatergic afferences (stim). (**B**) The frequency of GABAergic miniature inhibitory postsynaptic currents (mIPSC) in LA PNs was significantly reduced in Cln3$^{\Delta ex1-6}$ mice, whereas mean amplitude was unchanged (wt: $n = 27/7$ [number of recordings/number of mice]; $Cln3^{\Delta ex1-6}$: $n = 32/7$; Student's t-test; scale bars 100 pA and 50 ms). (**C**) The frequency and the amplitude of spontaneous (s) IPSC were reduced in Cln3$^{\Delta ex1-6}$ mice (wt: $n = 33/7$; $Cln3^{\Delta ex1-6}$: $n = 33/7$; Student's t-test). (**D**) After extracellular microstimulation of GABAergic afferents, peak amplitude of evoked IPSC in amygdala PNs was reduced (wt: $n = 25/7$; $Cln3^{\Delta ex1-6}$: $n = 28/7$; Student's t-test). (**E**) The frequency of miniature excitatory postsynaptic currents (mESPC) was reduced in $Cln3^{\Delta ex1-6}$ mice (wt: $n = 23/6$; $Cln3^{\Delta ex1-6}$: $n = 27/7$; Student's t-test; scale bars 20 pA and 500 ms). (**F**) The frequency of the spontaneous (s) EPSC was reduced while the mean amplitude was unaffected (wt: $n = 27/6$; $Cln3^{\Delta ex1-6}$; $n = 33/7$; Student's t-test; scale bars 20 pA and 500 ms). (**G**) The amplitude of evoked (e) EPSC was reduced in $Cln3^{\Delta ex1-6}$ mice (wt: $n = 23/6$; $Cln3^{\Delta ex1-6}$: $n = 24/7$; Student's t-test; scale bars 100 pA and 500 ms). Exact values of N, dispersion and precision measures are provided in *Supplementary file 2*.

DOI: https://doi.org/10.7554/eLife.28685.004

The following figure supplements are available for figure 2:

**Figure supplement 1.** In-out curve showing reduced eIPSC in the amygdala of $Cln3^{\Delta ex1-6}$ mice at the age of 14 months.
DOI: https://doi.org/10.7554/eLife.28685.005

**Figure supplement 2.** In-out curve showing reduced eEPSC in the amygdala of 14-month-old $Cln3^{\Delta ex1-6}$ mice.
DOI: https://doi.org/10.7554/eLife.28685.006

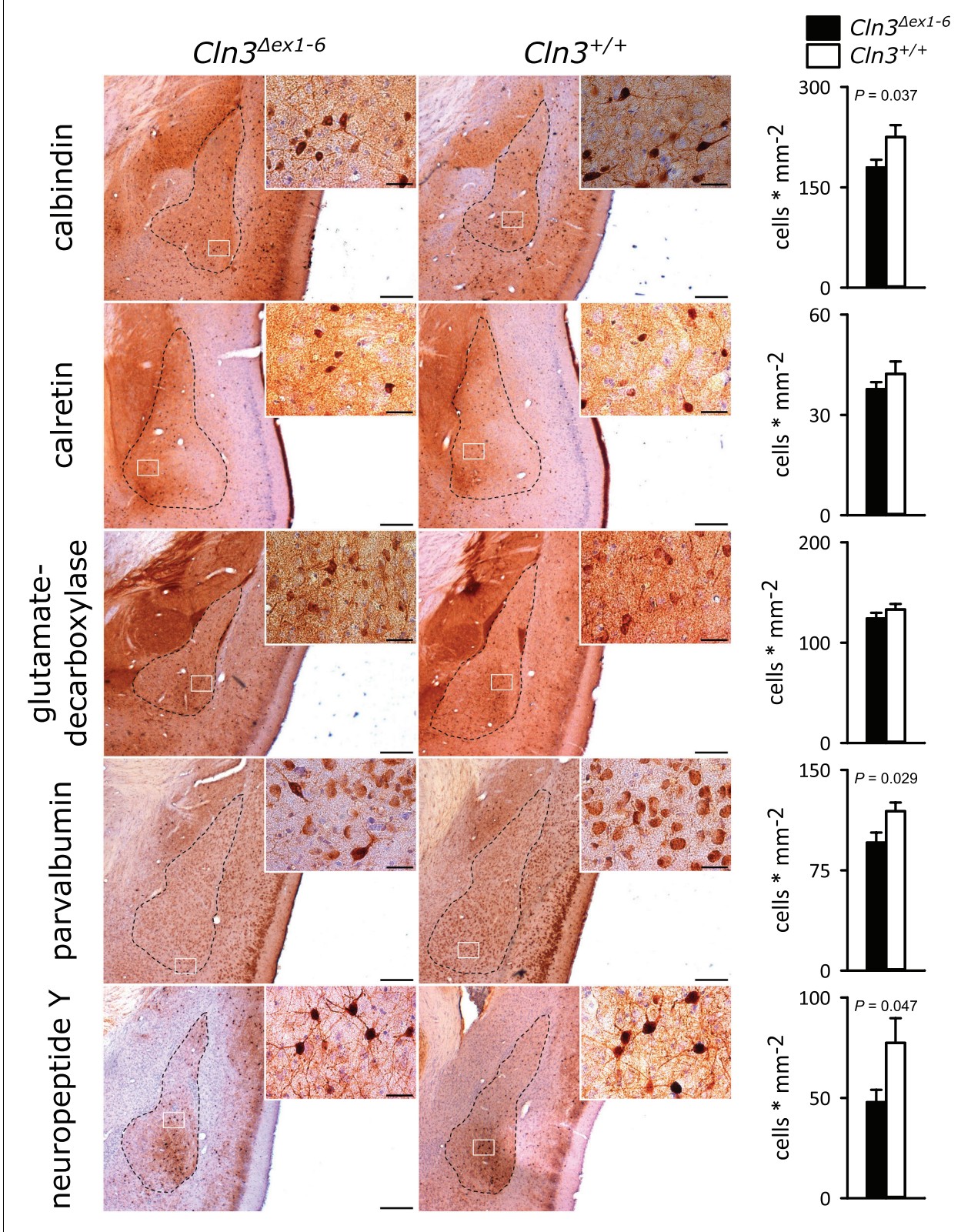

**Figure 3.** Loss of GABAergic interneurons in the amygdala of $Cln3^{\Delta ex1-6}$ mice at the age of 14 months. In coronar brain slices the area of the amygdala core regions was morphologically identified by their characteristic shape and distribution of neuronal nuclei (dashed lines). Interneuron subtypes were differentiated using immunohistological stains for the respective antigens. Note that in 14-month-old $Cln3^{\Delta ex1-6}$ mice the number of calbindin, parvalbumin, and neuropeptide Y positive interneurons in the amygdala area was reduced compared to age-matched wt littermates, whereas other

*Figure 3 continued on next page*

*Figure 3 continued*

markers were unchanged (scale bar: 500 μm; inset: 50 μm; *n = 13 to 16*, Mann-Whitney test). Exact values of N, dispersion and precision measures are provided in ***Supplementary file 2***.

DOI: https://doi.org/10.7554/eLife.28685.007

The following figure supplement is available for figure 3:

**Figure supplement 1.** Density of NeuN-positive cells is unchanged in the amygdala of 14-month-old Cln3$^{\Delta ex1-6}$ mice (*n = 9 vs. 8*).

DOI: https://doi.org/10.7554/eLife.28685.008

amygdala and hippocampus - associated learning deficits and to evaluate the underlying pathomechanisms, we used a standardized fear conditioning paradigm with 100% reinforcement in Cln3$^{\Delta ex1-6}$ mice at the age of 14 months (***Figure 4A***) and performed ex-vivo neurophysiological and quantitative stereological investigations in hippocampal slices. Recall of cued and contextual fear memory was significantly reduced in Cln3$^{\Delta ex1-6}$ mice on day 1 and day 2 after conditioning. The reduction of cued fear memory was still apparent after 1 week (***Figure 4B***). Based on our findings in the amygdala, we hypothesized that defective synaptic activity and neuronal dysfunction in the hippocampus might contribute to learning deficits of Cln3$^{\Delta ex1-6}$ mice. We therefore analyzed spontaneous and evoked GABAergic and glutamatergic synaptic transmission in granule cells of the dentate gyrus (***Figure 4C***). Again, membrane potential and input resistance were unchanged in both genotypes (membrane potential −71 ± 1.2 mV vs. −72 ± 1.0 mV; input resistance 192 ± 18 MΩ vs. 176 ± 16 MΩ; n = 17/6 vs. 14/6 [number of recordings/number of mice]; Cln3$^{\Delta ex1-6}$ mice vs. wt mice). GABAergic mIPSCs in presence of 1 μM TTX had reduced peak amplitudes in Cln3$^{\Delta ex1-6}$ mice (***Figure 4D***). Frequency and amplitude of sIPSCs were reduced in Cln3$^{\Delta ex1-6}$ mice as compared to wt littermates (***Figure 4E***). Peak amplitude of evoked GABAergic synaptic currents was also reduced in both conditions when stimulation was performed in the outer molecular layer (OML, stimulation of afferents arising from hilar and molecular layer perforant-path-associated interneurons) and in the granule cell layer (GCL; afferents primarily from local basket cells) indicating that GABAergic disinhibition is not restricted to specific hippocampal pathways or location of the postsynaptic receptor field in the GC (***Figure 4C and F***). Moreover, we found impaired paired pulse depression (PPD) in these pathways (***Kobayashi and Buckmaster, 2003***) indicating a defect in short-term plasticity (***Figure 4G***) and pointing toward impairment of presynaptic release mechanisms in hippocampal GABAergic interneurons.

In line with the findings in the amygdala, the frequency of the mEPSCs and sEPSCs was reduced in Cln3$^{\Delta ex1-6}$ mice while the amplitude was unaffected (***Figure 4H and I***). Corroborating these findings, eEPSCs were unchanged at minimal stimulation but peak amplitudes were severely affected in Cln3$^{\Delta ex1-6}$ mice at higher stimulation intensities of the lateral perforant path (LPP) (***Figure 4J***, ***Figure 4—figure supplement 1***). In this pathway, paired-pulse stimulation with an interstimulus interval of 100 ms usually would have resulted in facilitation. However, in Cln3$^{\Delta ex1-6}$ mice short-term plasticity of eEPSC was reduced (***Figure 4K***).

Similar to the morphometric analyses in the amygdala detailed stereological analysis of the hippocampus revealed loss of NPY$^+$ and Parv$^+$ interneurons in Cln3$^{\Delta ex1-6}$ mice with a slightly more pronounced reduction in the CA3 subfield of the hippocampus. GAD-positive interneurons were also reduced reflecting thorough reduction of inhibitory GABAergic interneurons in the hippocampus (***Figure 5*** and ***Figure 5—figure supplement 1***). Against this, the density of NeuN-positive cells within the principle cell layers of the dentate gyrus and the CA3 and CA1 subregions was not reduced (***Figure 5—figure supplement 2***).

Collectively, these findings suggest that defective synaptic transmission within the amygdala-hippocampal formation and loss of GABAergic interneurons contribute to the age-dependent anxious behavior and learning and memory impairment of Cln3$^{\Delta ex1-6}$ mice.

## Cerebellar dysfunction underlying motor deficits in cln3$^{\Delta ex1-6}$ mice

Next, we investigated motor function in the Cln3$^{\Delta ex1-6}$ mouse model. JNCL patients develop motor dysfunction including bradykinesia and ataxia which ultimately leads to loss of the ability to walk (***Schulz et al., 2013***). In Cln3$^{\Delta ex1-6}$ mice, motor deficits became apparent in a standardized rope climbing test at an age of 7 months and progressed up to the age of 14 months (***Figure 6A and B***,

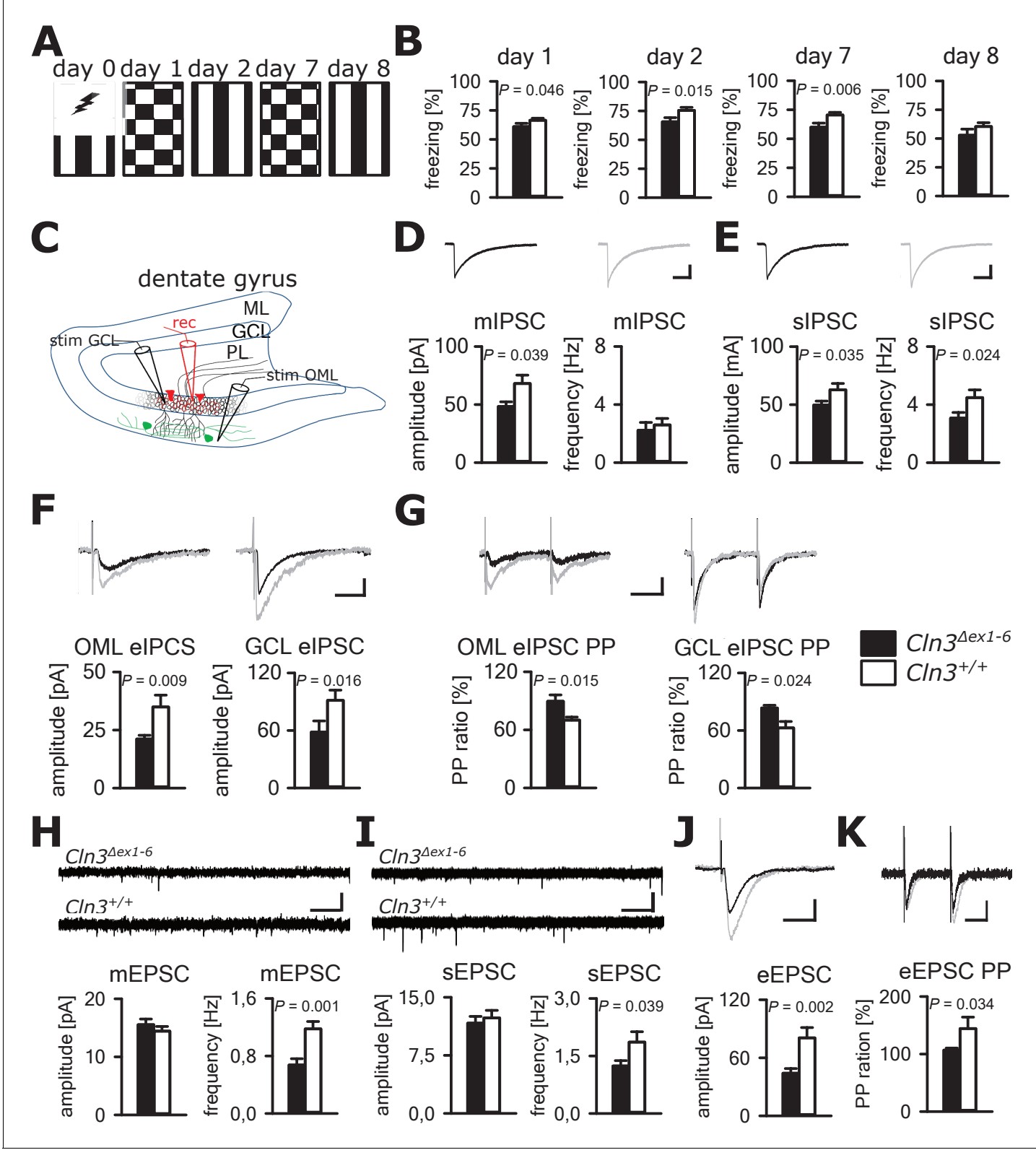

**Figure 4.** Learning deficits and disturbed hippocampal synaptic transmission in *Cln3*<sup>Δex1-6</sup> mice at the age of 14 months. (**A**) For induction of fear conditioned and contextual learning a conditioned acoustic stimulus (tone) was paired with an unconditioned stimulus (electric shock) on day 0 in a specific environment (schematically illustrated as black and white stripes). On days 1 and 7, the conditioned stimulus was presented in a 'novel´ environment (chequered pattern) to test conditioned fear learning. Contextual fear learning was tested on days 2 and 8 by presenting the same

*Figure 4 continued on next page*

*Figure 4 continued*

environment as on day 0 without the conditioned stimulus (tone). (B) Rehearsal of cued and context learning was impaired in *Cln3*$^{\Delta ex1-6}$ mice. Reduced freezing of *Cln3*$^{\Delta ex1-6}$ mice was detected on day 1 and day 7 (auditory cue in novel environment) and day 2 (conditioned environment without auditory cue) (n = 18 vs.23; Student's t-test). On day 8 both groups showed a reduction of contextual fear behavior without significant differences. (C) Schematic illustration of the patch-clamp recording situation in the hippocampal dentate gyrus (rec, shown in red). Stimulation electrodes (stim) were positioned in the granule cell layer (GCL) and outer part of the molecular layer (OML) to stimulate GABAergic efferents of different interneuron subtypes (PL: polymorphic layer). For recording of evoked excitatory postsynaptic currents stimulation was performed in the OML to stimulate the lateral perforant pathway (LPP). (D) Analysis of GABAergic mIPSC in presence of TTX revealed reduction of the peak amplitude, whereas the frequency was unchanged (example traces from a single neuron are shown in the upper panel; in grey: traces of a wt mouse, in black: traces of a *Cln3*$^{\Delta ex1-6}$ mouse; wt: n = 11 cells from nine mice; *Cln3*$^{\Delta ex1-6}$: n = 10/8; Student's t-test; scale bar: 10 ms / 20 pA). (E) Frequency and peak amplitude of sIPSC were significantly reduced in dentate gyrus GC of *Cln3*$^{\Delta ex1-6}$ mice; wt: n = 14/11; *Cln3*$^{\Delta ex1-6}$: n = 14/11; amplitude: Student's t-test; frequency: Mann-Whitney test; scale bar: 10 ms / 20 pA). (F) Minimal threshold stimulation of afferents was performed in the OML and the GCL to obtain eIPSC of individual afferents projecting from different hippocampal areas. In slices of *Cln3*$^{\Delta ex1-6}$ mice eIPSC derived from both stimulation sites showed reduced peak amplitude in comparison to wt littermates (OML: n = 11/11 vs. 11/11, GCL: n = 12/8 vs. 10/7; Student's t-test; scale bar: 25 ms / 10 pA). (G) Paired pulse depression as a characteristic feature of presynaptic short-term plasticity in the synapses on dentate gyrus GC was affected in *Cln3*$^{\Delta ex1-6}$ mice at both stimulation sites (OML: n = 10/10 vs. 11/11; GCL: n = 12/8 vs. 8/7; Student's t-test; scale bar: 50 ms / 10 pA). (H and I) miniature (m) EPSC frequency (H, wt: n = 15/7 vs. *Cln3*$^{\Delta ex1-6}$: n = 11/6; amplitude: Mann-Whitney test; frequency: Student's t-test) and spontaneous (s) EPSC frequency was reduced in the dentate gyrus GC in *Cln3*$^{\Delta ex1-6}$ mice (I, wt: n = 14/7; *Cln3*$^{\Delta ex1-6}$: n = 17/6; Mann-Whitney test; scale bars 20 pA and 500 ms). (J) The amplitude of eEPSC was reduced at high stimulation intensities (wt: n = 12/7; *Cln3*$^{\Delta ex1-6}$: n = 12/6; Student's t-test; scale bars 40 pA and 50 ms). (K) The paired-pulse facilitation of the LPP eEPSC at 100 ms interstimulus interval was reduced in *Cln3*$^{\Delta ex1-6}$ mice (wt: n = 12/7; *Cln3*$^{\Delta ex1-6}$: n = 12/6; Student's t-test; scale bars 50 pA and 50 ms). Exact p values, dispersion and precision measures are provided in ***Supplementary file 2***.

DOI: https://doi.org/10.7554/eLife.28685.009

The following figure supplement is available for figure 4:

**Figure supplement 1.** Reduced eEPSC amplitudes in the hippocampus of 14 months old *Cln3*$^{\Delta ex1-6}$ mice.

DOI: https://doi.org/10.7554/eLife.28685.010

*Video 1*). Muscle strength was not affected as measured by grip strength testing (*Figure 6—figure supplement 1*). Similarly, stride length and hind limb angle during normal movement on a plane surface revealed no differences at 7 and 14 months (*Figure 6—figure supplement 2*), thus suggesting intact lower motor neuron function. Therefore, impaired limb coordination may be the predominant cause of the deficits in climbing performance of *Cln3*$^{\Delta ex1-6}$ mice. Accordingly, the mean score for coordinated movement was affected in *Cln3*$^{\Delta ex1-6}$ mice (*Figure 6C*, left) and coordinative performance assessed on a semi-quantitative scale during the rope climbing correlated with the time to reach the platform (*Figure 6C*, right). At 14 months, the walking ability of *Cln3*$^{\Delta ex1-6}$ mice was also reduced as seen in the accelerating RotaRod test (*Figure 6D*) and in the OF test as determined by walking distance (*Figure 6E*) while both test revealed no disturbances in younger mice (*Figure 6D* and *Figure 1—figure supplement 1*). Regulation of movement coordination is mainly controlled by cerebellar function, whereas exact spinal integration is important for movement execution of the limbs. We therefore analyzed first cerebellar PC function since correct PC firing is the only output of information arising from the cerebellar cortex. Further, we tested spinal synaptic transmission and presynaptic inhibition as important pathways for spinal movement control.

Spontaneous PC action potential firing was measured in loose-patch configuration in acute cerebellar slices of *Cln3*$^{\Delta ex1-6}$ mice and wt littermates (*Figure 7A*). The interspike interval of action potential firing of PC was severely decreased (*Figure 7B*, left and C), whereas the regularity of spontaneous PC firing was not compromised (*Figure 7B*, right and C). Next, we tested PC firing after stimulation of parallel fibers (schematically shown in *Figure 7A*). The standard deviation of the latency of the first action potential after parallel fiber stimulation was significantly increased in *Cln3*$^{\Delta ex1-6}$ mice (*Figure 7D*). Further, the subsequent spiking pause was impaired in recordings from *Cln3*$^{\Delta ex1-6}$ mice as shown in the raster plot and peri-stimulus time histogram (*Figure 7E*). Corroborating these functional defects, detailed stereological analysis of interneurons in the cerebellar cortex of the vermis revealed loss of GAD65-positive interneurons in the molecular layer, whereas the number of Parvalbumin-positive interneurons and PCs as well as the total number of neurons was unchanged (*Figure 8*). Together, these observations are in good accordance with disturbed feedforward inhibition by molecular layer interneurons onto PC (*Wulff et al., 2009*).

In contrast, no abnormalities were seen in GABAergic spinal inhibition of *Cln3*$^{\Delta ex1-6}$ mice. The frequency-dependent depression of the spinal H-Reflex relies mainly on segmental GABAergic

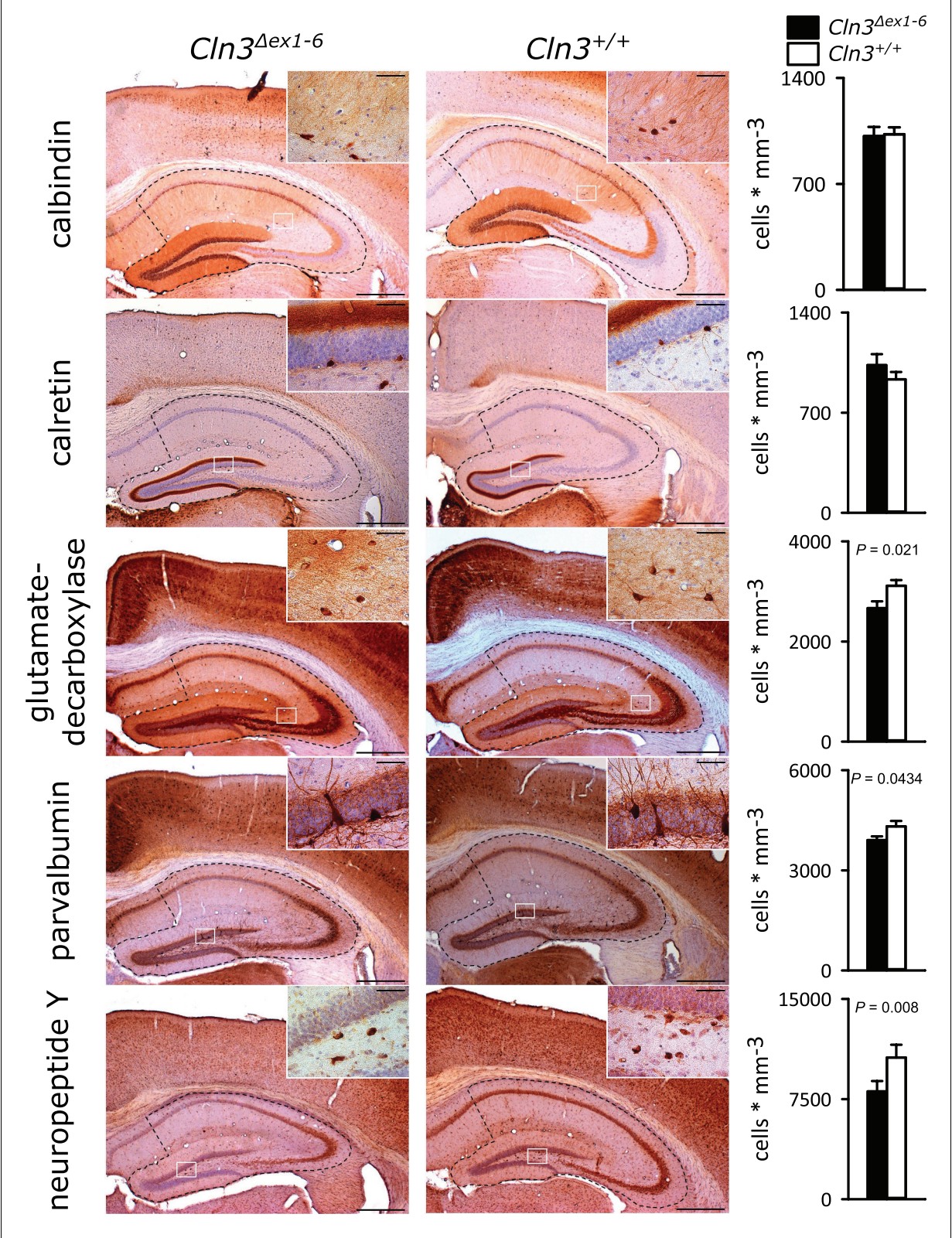

**Figure 5.** Loss of GABAergic interneurons in the hippocampus of $Cln3^{\Delta ex1-6}$ mice at the age of 14 months. Quantitative stereological analysis of interneuron subclasses revealed a reduced number of parvalbumin, neuropeptide Y and GAD-positive interneurons, whereas other markers were unchanged. Note that GAD-positive cells were not counted in the hilus of the dentate gyrus since here intense staining of the fiber network precluded
*Figure 5 continued on next page*

*Figure 5 continued*

precise cell differentiation (scale bar: 500 µm; inset: 50 µm; *n = 13 to 16*)). Exact values of N, dispersion and precision measures are provided in *Supplementary file 2*.

DOI: https://doi.org/10.7554/eLife.28685.011

The following figure supplements are available for figure 5:

**Figure supplement 1.** Density of interneurons within the hippocampal subfields of *Cln3^Δex1-6* mice and wt littermates.

DOI: https://doi.org/10.7554/eLife.28685.012

**Figure supplement 2.** Density of NeuN-positive cells is unchanged in the hippocampus of 14-month-old *Cln3^Δex1-6* mice.

DOI: https://doi.org/10.7554/eLife.28685.013

interneurons projecting to Ia afferent fibers. At time points of 7 and 14 months, the frequency-dependent depression of the spinal H-Reflex was not altered in *Cln3^Δex1-6* mice (*Figure 9A and B*). According to these findings, the amplitude and the time-dependent summation of the dorsal root potentials which directly reflect GABA-dependent presynaptic inhibition of primary afferent fibers

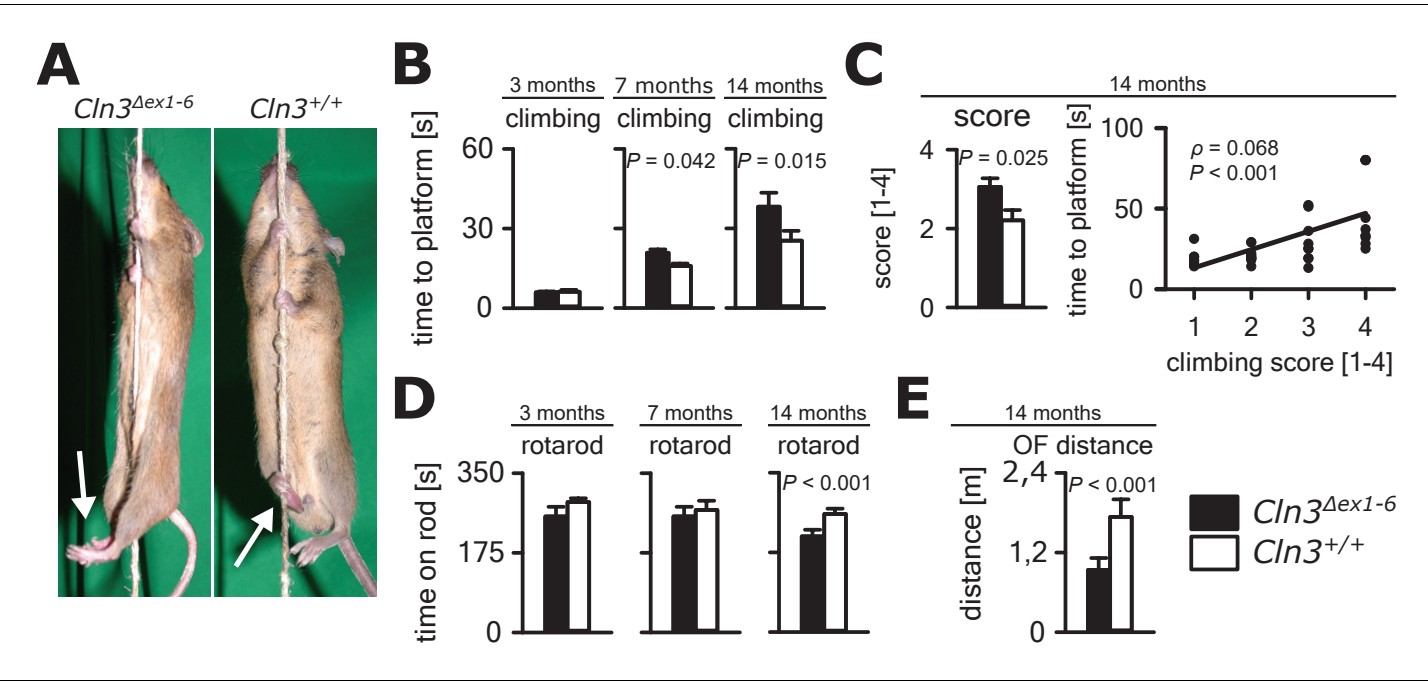

**Figure 6.** Progressive ataxia and motor impairment in *Cln3^Δex1-6* mice. (A, B) Mice were trained to climb a vertically stretched cord of 1 m length to reach a platform (climbing test). Mice used their limbs in an alternating manner that allows fast movement and rapid climbing (see also *Video 1*). Time to platform of 3 months old *Cln3^Δex1-6* mice was unchanged to wt littermates. At later time points, *Cln3^Δex1-6* mice developed an abnormal climbing pattern leading to prolonged climbing time at the age of 7 months (*n = 17 vs. 18*, Mann-Whitney test) and more pronounced at an age of 14 months (*n = 17 vs. 19*, Mann-Whitney test). Moreover, *Cln3^Δex1-6* mice showed ataxic limb movements that led to difficulties in grasping the rope (arrows in A). (C) The coordination of hind limb movement was significantly impaired in *Cln3^Δex1-6* mice at 14 months as assessed by the climbing score evaluating hind limb use and correlated with the time needed to reach the platform (*n = 17 vs. 19*, Mann-Whitney test; Spearman's rank correlation coefficient $\rho = 0.68$). (D) Motor performance tested on the RotaRod revealed deficits of forced walking behavior in *Cln3^Δex1-6* mice at 14 months when compared to age-matched wt littermates (3 months: *n = 12 vs. 18*; 7 months: *n = 13 vs. 18*, both Mann-Whitney test; 14 months: *n = 17 vs. 21*, Student's t-test). (E) In the OF the total walking distance was reduced in 14 month old *Cln3^Δex1-6* mice (*n = 18 vs. 20*, Mann-Whitney test). Exact dispersion and precision measures are provided in *Supplementary file 2*.

DOI: https://doi.org/10.7554/eLife.28685.014

The following figure supplements are available for figure 6:

**Figure supplement 1.** Grip strength of *Cln3^Δex1-6* mice and wt littermates at different ages.

DOI: https://doi.org/10.7554/eLife.28685.015

**Figure supplement 2.** Gait analysis of *Cln3^Δex1-6* mice and wt littermates.

DOI: https://doi.org/10.7554/eLife.28685.016

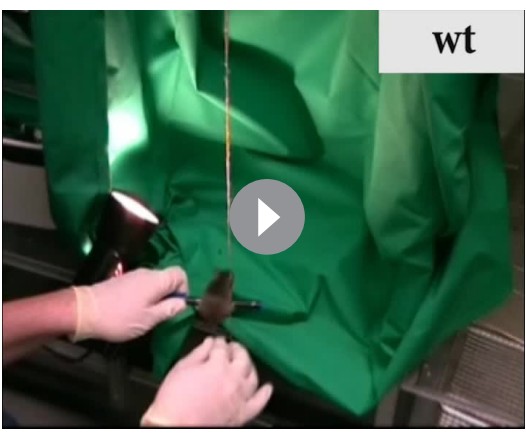

**Video 1.** Rope climbing performance of a wt littermate and a $Cln3^{\Delta ex1-6}$ mouse. The wt mouse notices and grasps the vertically spanned rope quickly and climbs up the rope using all four limbs in an alternating manner. The $Cln3^{\Delta ex1-6}$ mouse notices and grasps the rope but is severely disabled to climb up the rope properly. It can easily be observed that the $Cln3^{\Delta ex1-6}$ mouse is not able to properly set the hind limbs on the rope and to use the hindlimbs for climbing. Instead hind limbs move in an ataxic pattern clasping in an uncoordinated way (climbing score 4).
DOI: https://doi.org/10.7554/eLife.28685.017

was not affected in $Cln3^{\Delta ex1-6}$ mice (**Figure 9C and D**). Corroborating these findings, the amount of spinal GABAergic interneurons was not reduced (**Figure 9—figure supplement 1**).

## Discussion

We used the $Cln3^{\Delta ex1-6}$ mouse model of JNCL to investigate functional and morphological changes in several CNS networks that may reflect crucial pathophysiological mechanisms of this fatal inherited CNS disorder of children and young adults. Our principal finding is that GABAergic and glutamatergic synaptic transmission is critically affected in supraspinal centers leading to heightened anxiety, neurocognitive dysfunction, and defects in motor coordination. We suggest that these syndromatic features of this complex disease are largely caused by synaptic dysfunction and disturbed presynaptic release properties and by loss of distinct classes of interneurons.

### Reduced learning abilities and increased anxiety behavior in $Cln3^{\Delta ex1-6}$ mice is based on defective synaptic transmission

We put special emphasis on the analysis of anxiety behavior and learning deficits in the $Cln3^{\Delta ex1-6}$ mice, since comorbidity of affective disorder and severe cognitive abnormalities is common in JNCL patients (**Santavuori et al., 1993**). Due to the progressive cognitive impairment, the assessment and quantification of other psychiatric abnormalities in JNCL patients including anxiety is difficult and may therefore be largely underestimated (**Bäckman et al., 2001**; **Bäckman et al., 2005**; **Adams et al., 2007**). In the $Cln3^{\Delta ex1-6}$ mouse model, we here identified several abnormalities that may indeed mimic abnormal features in patients with JNCL. Specifically, $Cln3^{\Delta ex1-6}$ mice developed a progressive decrease of explorative behavior over lifetime and a significant increase in anxiety-like behavior using complementary test paradigms at the age of 7 and 14 months. Moreover, in line with previous experiments using a light-cued T-maze (**Wendt et al., 2005**), we observed that $Cln3^{\Delta ex1-6}$ mice develop deficits in contextual fear learning and cued fear memory. As the amygdala-hippocampal complex is of particular importance in memory formation and in fear processing (**Tovote et al., 2015**; **Kesner, 2013**), these deficits are best explained by the severely afflicted synaptic transmission at late time points which we found in the hippocampus and the amygdala.

Disturbed synaptic transmission and impaired short-term plasticity are in line with the proposed dysfunction of CLN3 protein, along with the neurodegenerative processes in patients and in the respective mouse models. CLN3 is localized in synaptosomes (**Luiro et al., 2001**), in axonal compartments, and in synaptic spines (**Oetjen et al., 2016**). The proposed role of CLN3 in endocytosis and endosomal functions provides a direct link to short-term plasticity and suggests a role of CLN3 in neuronal transmission. (**Luiro et al., 2004**; **Lojewski et al., 2014**; **Watanabe et al., 2014**; **Lou et al., 2012**). Moreover, sorting and correct membrane-insertion of AMPA- and GABA$_A$ receptors is regulated by endosomal pathways (**Park et al., 2004**; **Jacob et al., 2008**) and these processes may not function because of loss or dysfunction of CLN3. Furthermore, dysfunctional CLN3 may lead to deficits in spine morphology and to a reduced number of active synapses (**Golabek et al., 2000**; **Esteves da Silva et al., 2015**; **Padamsey et al., 2017**). In addition, age-dependent changes in axonal excitability have been described in the $Cln3^{\Delta ex7/8}$ mouse model (**Burkovetskaya et al., 2017**). Different from our study, synaptic transmission was not evaluated directly by single-cell recordings in this study. Increase in synaptic strength was measured by extracellular recordings and was discussed

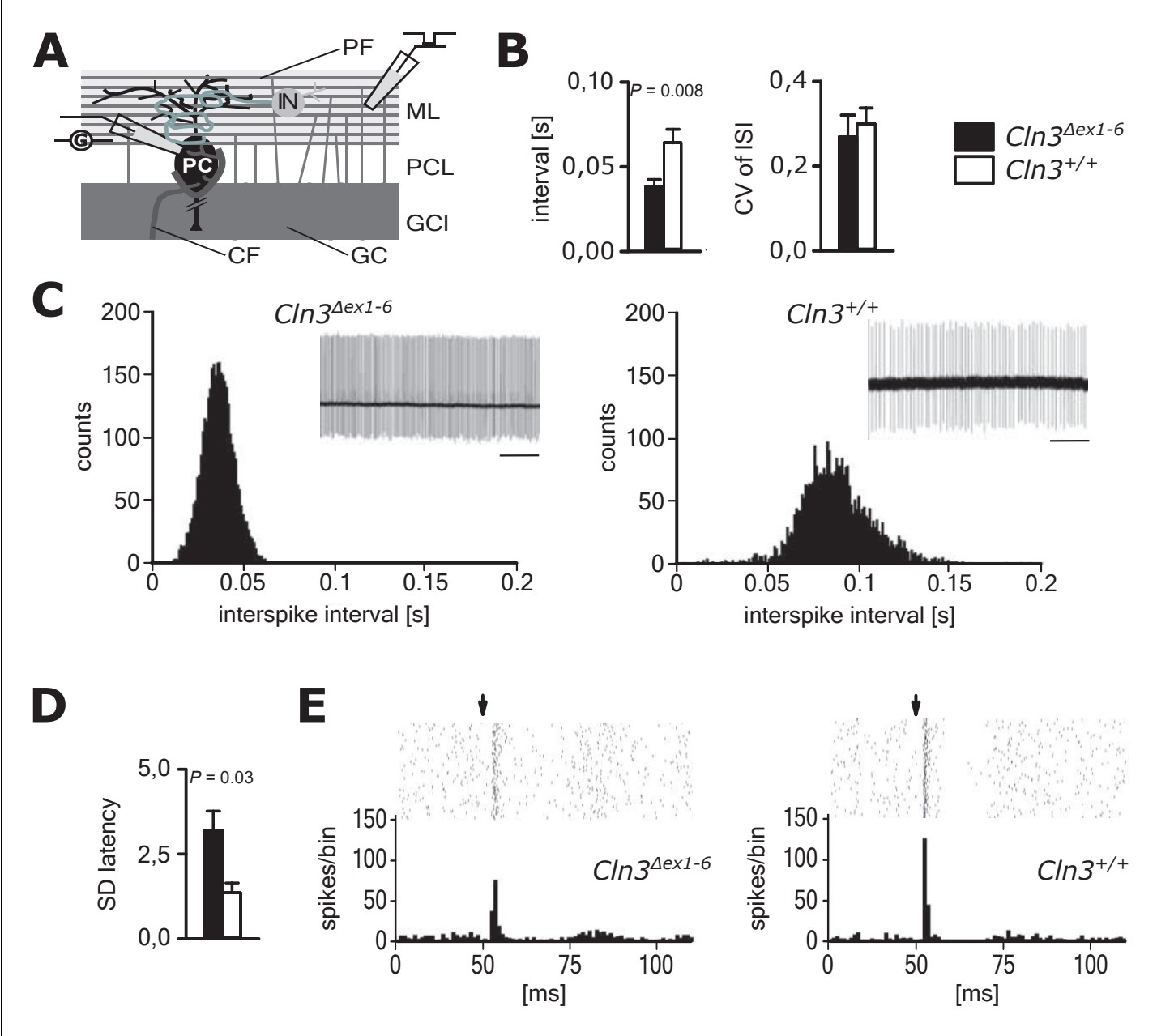

**Figure 7.** Pathological firing of cerebellar Purkinje cells (PC) of *Cln3^{Δex1-6}* mice at the age of 14 months. (**A**) Schematic drawing of neurons and connections in the cerebellar cortex. Spontaneous and evoked action potentials of PC were recorded in loose patch configuration. Granule cells (GC) give rise to the parallel fibers (PF) which excite PC and local interneurons (IN). IN induce feed forward inhibition onto PC (CF = climbing fiber; GCL = granule cell layer; ML = molecular layer; PCL = Purkinje cell layer). (**B**) Inter-spike intervals of spontaneous action potential firing were reduced in *Cln3^{Δex1-6}* mice while the regularity of spiking (CV ISI) was unchanged (n = 12 cells from 8 mice vs. 12/6, Student's t-test). (**C**) Histograms of inter-spike intervals of spontaneous activity and representative recordings (upper right inset, scale bar: 500 ms) are shown. Inter-spike intervals were reduced and shifted to smaller bin sizes in *Cln3^{Δex1-6}* mice. (**D**) The standard deviation of the latency of the first action potential after parallel fiber stimulation was increased in *Cln3^{Δex1-6}* indicating enhanced jitter of PC firing upon PF activation (n = 12/8 vs. 13/6, Mann-Whitney test). (**E**) Representative raster plots and histograms showing PC simple spikes upon single stimulation of parallel fibers. Each horizontal series of points in the raster plot represents an individual recording. Arrows indicate onset of stimulation after baseline recording of 50 ms. In the example of a wt mouse stimulation elicited simple spikes in a narrow time window and a clear spiking pause following the initial response, while in the *Cln3^{Δex1-6}* mouse the jitter of firing after PF stimulation is increased and the spiking pause is reversed. Exact dispersion and precision measures are provided in *Supplementary file 2*.
DOI: https://doi.org/10.7554/eLife.28685.018

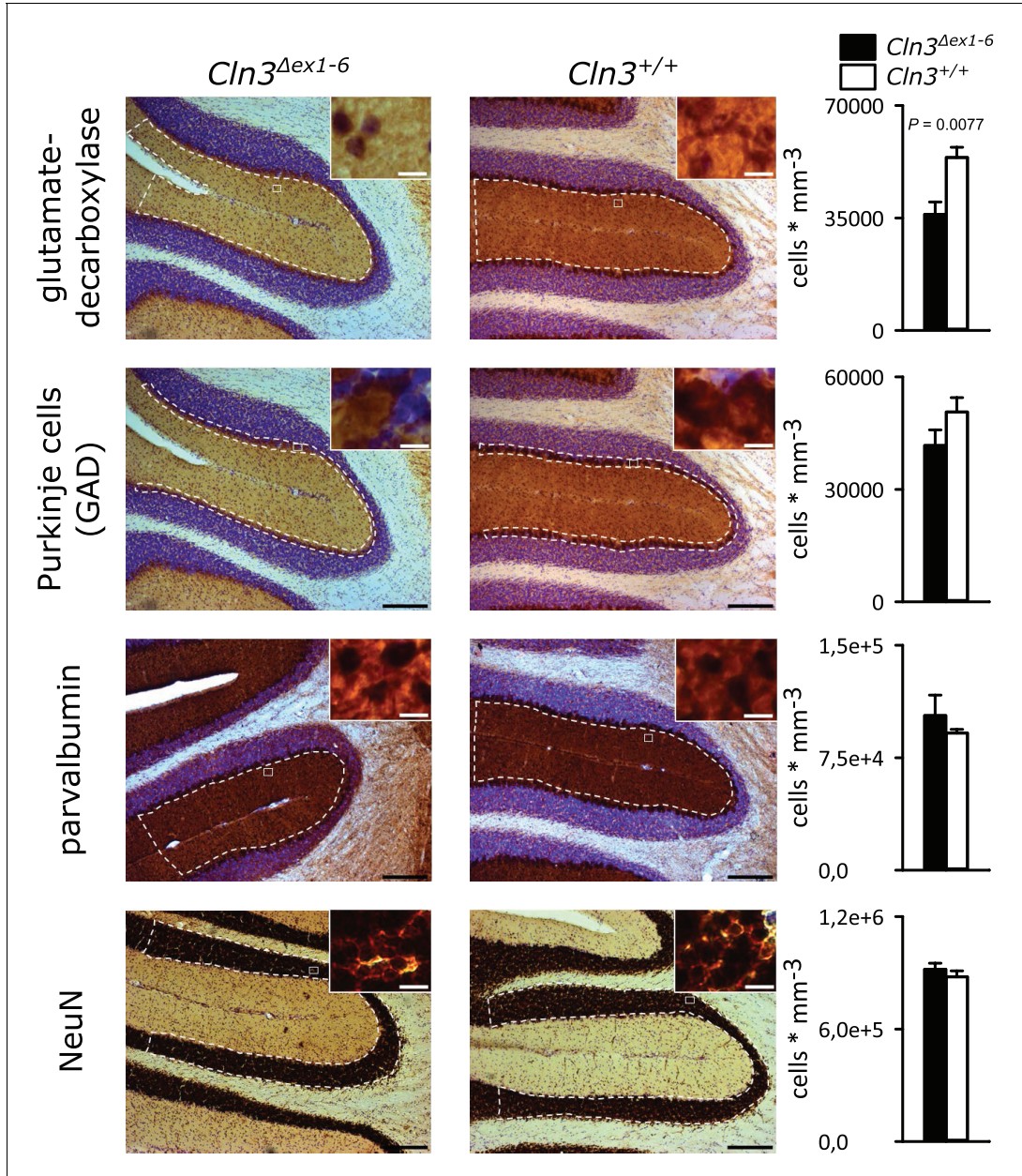

**Figure 8.** Reduction of GAD-positive interneurons in the cerebellar molecular layer of 14 months old *Cln3^Δex1-6* mice. Quantitative stereological analysis of interneuron subclasses and NeuN-positive neurons in coronar cerebellar slices revealed a reduced number of GAD-positive interneurons within the molecular layer (ML) (n = 9 vs. 8, Student's t-test) whereas other markers as well as the number of Purkinje cells were unchanged (scale bar: 200 μm; inset: 10 μm). Exact values of N, dispersion and precision measures are provided in *Supplementary file 2*.

DOI: https://doi.org/10.7554/eLife.28685.019

as a compensatory mechanism to keep neuronal activity in a normal range. Considering the precautions with respect to different animal models, diverse recording methods, and different neuronal networks that have been analyzed, we here propose that synaptic transmission deficits are causative factors of disease instead of compensatory regulatory mechanisms. This assumption is based on our more direct and extensive functional evaluations showing consistent results in several and independent systems in the CNS.

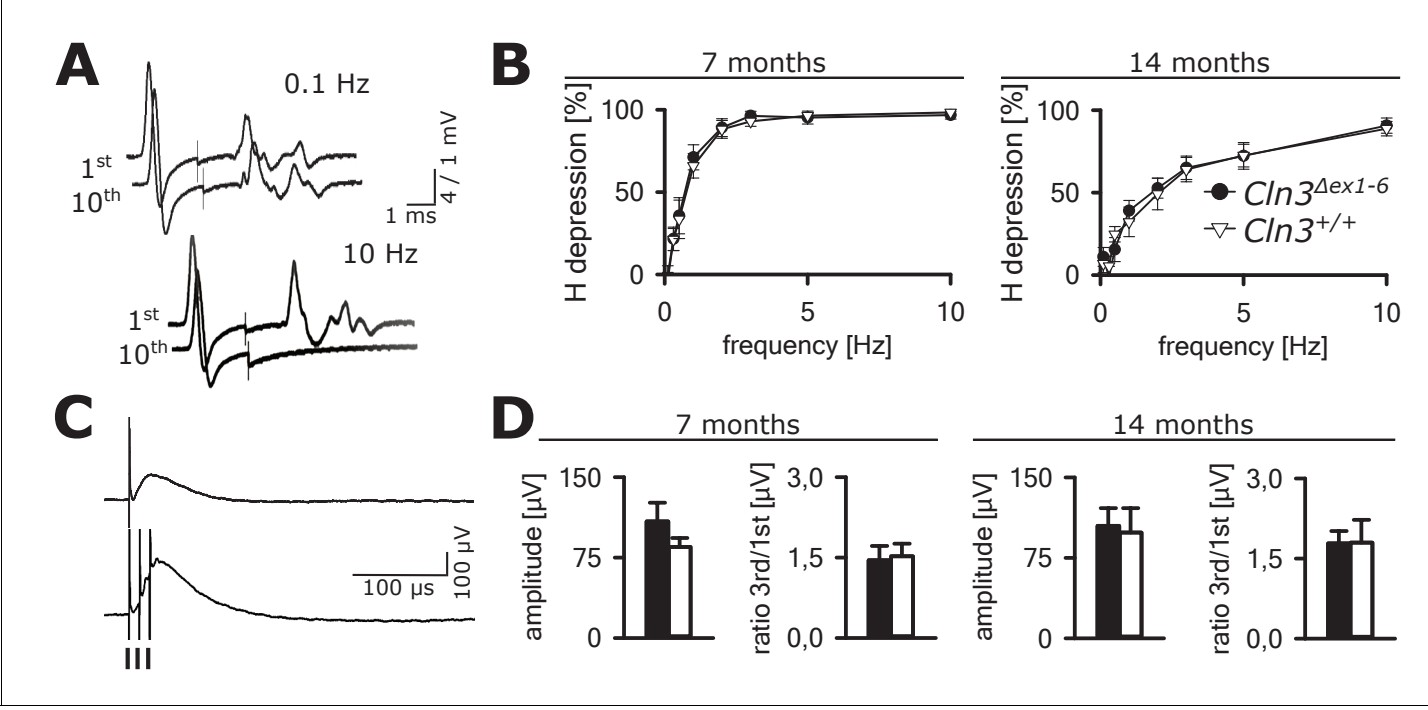

**Figure 9.** Spinal networks are unaffected in *Cln3^(Δex1-6)* mice. (**A**) Example traces for H-Reflex recordings in mice; the first major deflection is the motor response (M-wave) upon anterograde sciatic nerve stimulation, the second is the H-Reflex after monosynaptic transmission in the spinal cord followed by a polyphasic F-wave.1st and 10th trace of 10 consecutive stimuli are shown for the indicated frequencies. Note that H-Reflex amplification is fourfold higher than for the M-wave. The H-Reflex is unchanged after 10 serial stimuli at 0.1 Hz stimulation, whereas it is completely abolished (100% depression) after 10 stimuli at 10 Hz (post-activation depression, primarily mediated by local GABAergic interneurons). (**B**) Post-activation depression of the H-Reflex in *Cln3^(Δex1-6)* and wt mice was tested at the age of 7 and 14 months and was unchanged at both time points at all investigated stimulation frequencies. Note, that the depression of H-Reflex is shifted to higher frequencies in both genotypes at higher age of 14 months, indicating age-dependent changes within the spinal networks in both groups (7 months $n$ = 12 recordings from 10 mice vs. 14/13, 14 months $n$ = 16/10 vs. 13/8; Two-way ANOVA). (**C**) Example traces for recordings of dorsal root potentials (DRP) in mice in-vivo as a direct measurement of spinal GABAergic presynaptic inhibition. The upper trace shows a recording of the DRP after a single stimulus, the lower trace shows the DRP after three stimuli demonstrating temporal summation. Arrows indicate electric stimulations. (**D**) Peak amplitudes of DRP after a single stimulus and the ratio of peak amplitudes after time-dependent summation vs. single stimulation are not significantly different in *Cln3^(Δex1-6)* and wt mice at both analyzed time points indicating unchanged spinal presynaptic inhibition (7 months $n$ = 14 recordings from 8 mice vs. 17/8, 14 months $n$ = 17/9 vs. 10/6). Exact p values, dispersion and precision measures are provided in **Supplementary file 2**.

DOI: https://doi.org/10.7554/eLife.28685.020

The following figure supplement is available for figure 9:

**Figure supplement 1.** Number of spinal cord interneurons are unaffected.

DOI: https://doi.org/10.7554/eLife.28685.021

### Loss of interneurons in the amygdala and hippocampus directly contributes to defective GABAergic signaling and affects local networks *Cln3^(Δex1-6)* mice

In addition to the functional deficits, we found a partial loss of specific subclasses of interneurons in the amygdala and in the hippocampus while the overall number of neurons was unaltered in both regions thus suggesting a cell-type specific neuronal loss of GABAergic interneurons. We suggest that this neuronal loss may directly contribute to the abnormal GABAergic synaptic transmission in the amygdala and the hippocampus of *Cln3^(Δex1-6)* mice and thereby foster the imbalance of network inhibition and excitation. Deficits in the *Cln3^(Δex1-6)* mice as shown here resemble those of the GAD65 knockout mouse and other animal models with impaired inhibition in the amygdala (*Stork et al., 2003*; *Sangha et al., 2009*; *Lange et al., 2014*; *Geis et al., 2011*). The loss of distinct subgroups of interneurons in these regions may therefore at least partly account for disturbed GABAergic inhibition and heightened anxiety-like behavior. Parv$^+$ and Calb$^+$ cells constitute the vast majority of

interneurons in the basolateral amygdala (*Sah et al., 2003*). Thus, a reduction of Parv[+] and Calb[+] - interneurons as shown here likely contributes to decreased sIPSC frequency and reduced eIPSC amplitude in the PN of the amygdala. NPY[+] interneurons inhibit PNs in the amygdala (*Tasan et al., 2016*) and Calb[+] interneurons coordinate hippocampal-amygdala theta-oscillations (*Bienvenu et al., 2012*) known to be associated with retrieval of learned fear (*Seidenbecher et al., 2003*; *Pape et al., 2005*). Furthermore, hippocampal theta input to the amygdala shapes feed forward inhibition to gate heterosynaptic plasticity (*Bazelot et al., 2015*), which in turn seems critical for cued-specific signal integration (*Lange et al., 2014*). Indeed, in other mouse models, for example in Gad65$^{\Delta ex1-6}$ mice a reduction in theta oscillations is associated with deficits in retrieval of fear-related memory (*Mańko et al., 2012*; *Bergado-Acosta et al., 2008*; *Lange et al., 2014*). Parv[+] interneurons are involved in regulating PN activity during aversive associative learning, thereby controlling the strength of associative memory (*Wolff et al., 2014*). Thus, a reduction of these interneuron subclasses together with the compromised presynaptic function due to CLN3 dysfunction may lead to disinhibition of amygdala networks and finally be the pivotal mechanism explaining decreased cued learning performance and increased anxiety-levels in Cln3$^{\Delta ex1-6}$ mice (*Tovote et al., 2015*; *Nuss, 2015*).

In the hippocampus, we also found reduced numbers of Parv[+] and NPY[+] interneurons as well as a reduction of GAD[+] interneurons. The decreased eIPSC amplitude in GC of Cln3$^{\Delta ex1-6}$ mice after stimulation of the outer molecular layer may be a direct consequence of reduced NPY[+] interneurons. Large numbers of NPY receptor-positive synapses are found on dendrites of GC and a substantial population of the molecular layer perforant path-associated neurons are NPY[+] interneurons, whose axon collaterals are largely constrained to the outer and medial molecular layer (*Armstrong et al., 2011*; *Sperk et al., 2007*; *Hosp et al., 2014*). Concordantly, the reduced number of Parv[+] cells may directly contribute to the reduction of eIPSC after stimulation within the granule cell layer. The Parv[+] basket cells with their widespread axonal arborization in the GC layer innervate mainly the perisomatic parts of the GC (*Kraushaar and Jonas, 2000*) Loss of both Parv[+] and NPY[+] interneurons contributes to the reduction of spontaneous and evoked IPSC in dentate GC of Cln3$^{\Delta ex1-6}$ mice.

Parv[+] interneurons exert feedforward – and feedback inhibition in the hippocampus and may therefore be crucial for pattern separation and generation of network oscillation (*Hu et al., 2014*). In the dentate gyrus, this could contribute to neuronal hyperexcitability reducing the threshold for the development of temporal lobe seizures that often occur in JNCL patients (*Zhang and Buckmaster, 2009*). Selective loss of NPY[+] interneurons seems to precede cognitive impairment in a mouse models of accelerated senescence (*Sawano et al., 2015*) and might therefore be particularly detrimental in disease progression also in the Cln3$^{\Delta ex1-6}$ mouse model and JNCL patients. Recently, it has been shown that NPY itself is important for neuroprotection and modulation of synaptic plasticity, memory, anxiety-related behavior, and stress resilience (*Gøtzsche and Woldbye, 2016*; *Reichmann and Holzer, 2016*), all of which are core features in JNCL. Furthermore, there is compelling evidence that NPY acts as an endo- and exogenous anticonvulsant (*Erickson et al., 1996*; *Noè et al., 2008*). Thus, reduced levels of NPY[+], Parv[+], Calb[+] and GAD[+] interneurons in the amygdala-hippocampal complex may collectively contribute to a large number of key symptoms in Cln3$^{\Delta ex1-6}$ mice and JNCL. Taken together, we provide evidence that central disinhibition in Cln3$^{\Delta ex1-6}$ mice is likely caused by both synaptic dysfunction and loss of GABAergic interneurons. This may contribute to the imbalance of transmitter systems and may explain disturbed transmitter cycling and decreased amount of GABA as reported in previous studies (*Salek et al., 2011*; *Pears et al., 2005*). Once confirmed, this pathophysiological pathway might become a potential target for designing therapeutic strategies.

## Motor deficits and ataxia as a result of cerebellar disorganization

We here report age-dependent, specific coordinative deficits in the Cln3$^{\Delta ex1-6}$ mice extending previously reported findings on motor abnormalities (*Pontikis et al., 2004*; *Weimer et al., 2009*). Cln3$^{\Delta ex1-6}$ mice showed severe, age-dependent ataxia in the rope climbing test. It has been reported that Cln3$^{\Delta ex1-6}$ mice have reduced numbers of cerebellar granule cells and neurons in the deep cerebellar nuclei (*Haltia, 2003*; *Weimer et al., 2009*). We here found significantly enhanced PC simple spike firing and disturbed timing in Cln3$^{\Delta ex1-6}$ mice. Parallel fibers not only excite PCs but also the adjacent interneurons within the molecular layer of the cerebellar cortex which themselves have inhibitory connections onto the PCs accounting for a feed-forward inhibitory mechanism

(*Mittmann et al., 2005*; *Apps and Garwicz, 2005*). Upon parallel fiber stimulation, this inhibitory input onto the PCs follows the excitatory inputs from the parallel fibers at short intervals. The inhibitory inputs curtail the excitatory postsynaptic potentials (EPSPs) by short-circuiting the membrane through opening of GABA$_A$-receptor-associated chloride channels. This ensures exact time-dependent summation of incoming EPSPs leading to synchronized PC firing. The observed abnormalities with increased simple spike firing rate and loss of synchronization after parallel fiber stimulation in the *Cln3*$^{\Delta ex1-6}$ mice are best explained by a dysfunction of local inhibitory networks as seen in mice lacking GABA$_A$ receptors in PCs (*Wulff et al., 2009*). This hypothesis is further corroborated by the loss of GAD-positive interneurons in the molecular layer of the cerebellar cortex. Thus, similar to the changes in the amygdala-hippocampal complex, motor deficits in this mouse model of Batten disease are likely resulting from disturbed supraspinal cerebellar synaptic organization and GABAergic inhibition.

In contrast, we could not find evidence for defective synaptic transmission onto spinal motoneurons as determined by H-Reflex recording. In analogy to the hypothesis of disturbed synaptic transmission and GABAergic disinhibition at supraspinal centers, we additionally quantified spinal presynaptic inhibition, a potent spinal inhibitory mechanism mediated by local GABAergic interneurons (*Geis et al., 2010*; *Rudomin, 2009*). Consistent with our H-reflex findings, presynaptic inhibition was unaffected and quantitative analysis of interneuron subtypes revealed unchanged expression in the spinal cord.

Taken together, our study provides evidence for disturbed supraspinal synaptic transmission as a potential causative mechanism in the *Cln3*$^{\Delta ex1-6}$ mouse model of Batten disease. This is observed in several brain networks accounting for a major part of characteristic disease symptoms in patients with Batten disease. The changes in GABAergic transmission are likely mediated by dysfunction or loss of certain subtypes of GABAergic interneurons in the affected brain regions. JNCL patients and also the *Cln3*$^{\Delta ex1-6}$ mouse model of Batten disease harbor autoantibodies to GAD65 (*Chattopadhyay et al., 2002*). These autoantibodies are common also in autoimmune disorders of the CNS, for example stiff person syndrome and subtypes of limbic encephalitis or cerebellar ataxia (*Saiz et al., 2008*). According to the present knowledge, the role of these autoantibodies is not clear. However, a direct pathogenic role in these disorders and also in Batten disease appears questionable (*Werner et al., 2015*; *Dalmau et al., 2017*; *Gresa-Arribas et al., 2015*). Considering the specific loss of certain subtypes of interneurons shown here, the role of anti-GAD65 specific autoantibodies and T-cell reactivity to GAD65 should be investigated also in Batten disease.

# Materials and methods

## Key resources table

| Reagent type | Designation | Source or reference | Identifiers | Additional information |
|---|---|---|---|---|
| Strain, strain background (*Mus musculus*) | *Cln3*$^{\Delta ex1-6}$ mice | *Mitchison et al., 1999*. Neurobiol Dis, 10.1006/nbdi.1999.0267 | MGI:1933976 | Initial breeding pairs from: David A. Pearce Sanford Children's Health Research Center, Sanford Research, 2301 E 60th Street North, Sioux Falls, SD 57104, USA |
| Antibody | Anti-Calbindin | Swant, Marly, Switzerland; CB38 | RRID: AB_10000340 | Dilution: 1:5000, overnight at 4°C |
| Antibody | Anti-Calretinin | Swant, Marly, Switzerland; 7697 | RRID: AB_2619710 | Dilution: 1:5000, overnight at 4°C |
| Antibody | Anti-GAD65/67 | Merck Millipore, Darmstadt Germany; AB1511 | RRID: AB_90715 | Dilution: 1:8000, overnight at 4°C |
| Antibody | Anti-Neuropeptid Y (NPY) | Thermo Fisher Scientific, Waltham, USA; PA5-19568 | RRID: AB_10987237 | Dilution: 1:1000, overnight at 4°C |
| Antibody | Anti-Parvalbumin | Swant, Marly, Switzerland; PV27 | RRID: AB_2631173 | Dilution: 1:5000, overnight at 4°C |
| Antibody | Anti-Somatostatin | Acris Antibodies, Herford, Germany; AP01098SU-N | RRID: AB_1623011 | Dilution: 1:50, overnight at 4°C |

## Animals and experimental groups

All animal experiments were approved by the respective Bavarian and Thuringian state authorities (No. 55.5–2531.01-12/10; 78/05 and 02-44/12). All efforts were made to minimize animal suffering and to reduce the number of animals used. The study was performed in accordance with the ARRIVE guidelines for reporting animal research (*Kilkenny et al., 2010*). Breeding pairs of $Cln3^{\Delta ex1-6}$ mice on a 129/SvJ background were generated in the laboratory of D. Pearce (*Mitchison et al., 1999*). The $Cln3^{\Delta ex1-6}$ mouse model on the 129S6/SvJ background shows a similar and very pronounced phenotype in comparison to other currently available mouse models of JNCL, for example Cln3$^{\Delta ex7/8}$ mice (*Kovács and Pearce, 2015*). $Cln3^{\Delta ex1-6}$ mice and wt littermates were kept together in a 12 hr light-/dark cycle in standard cages with free access to water and food. Genotypes of $Cln3^{\Delta ex1-6}$ mice and wt littermates were determined using standard PCR methods and the following primer: CLN3E × 1 Fo (TGTATAGCAGACAGCG GACC), CLN3M6R (CAC TCC GAC TAT CCA ACC GA), NIHNeo3'Fo (TCG CCT TCT TGA CGA GTT CT) leading to gene products of ∼600 bp and ∼350 bp, for $Cln3^{\Delta ex1-6}$ and wt mice, respectively.

Altogether, 71 $Cln3^{\Delta ex1-6}$ mice and 73 wt littermates were used in the study. We used separate groups of $Cln3^{\Delta ex1-6}$ mice and wt littermates for each experimental procedure. For allocation of experimental animals in experimental groups see *Supplementary file 1*. Investigators performing behavioral tests, histological analyses, and electrophysiological studies were blinded as to the genotype of the mice.

For surgery and in vivo electrophysiological recordings animals were anesthetized using a xylazine-ketamine solution (8 mg/kg xylazine and 80 mg/kg body weight ketamine; Bayer, Leverkusen, Germany). For tissue harvesting, mice were deeply anaesthetized using isoflurane (Forene, AbbVie) and transcardially perfused with phosphate-buffered saline and freshly made 4% paraformaldehyde (Sigma Aldrich, Darmstadt, Germany). Tissue was harvested and immediately frozen in −37°C cold methyl butane and subsequently stored at −80°C. For electrophysiological in vitro slice recordings, animals were deeply anaesthetized and decapitated. Slices of the respective brain regions (hippocampus, amygdala, or cerebellum) were prepared using a VT1200 S Microtome (Leica, Wetzlar, Germany).

## Behavioral testing

$Cln3^{\Delta ex1-6}$ mice and wt littermates were tested for motor performance, coordinative function, muscle strength, anxiety-related behavior, and aversive learning in standardized behavioral testing procedures in a blinded manner. Behavioral testing was performed at the time points of 3, 7, and 14 months. Time points of behavioral testing are indicated in the Figures and Figure Legends for each test.

## RotaRod

Motor performance was tested with an accelerating RotaRod (TSE Systems Inc.,Bad Homburg, Germany ). Acceleration was set to increase from 5 to 30 rpm until testing month 7 and from 5 to 50 rpm for all later time points (accounting for the training effect in RotaRod testing over time). Initially, animals were trained on 5 consecutive days to obtain stable baseline values. During the experiment animals were tested monthly. The time staying on the rod was automatically measured and stored on a PC using TSE software. The median of the three best runs out of 5 trials on one day was used for later analyses. Data for the 3-month time point was obtained from a single experimental group, and data for the 7- and 14-month time points were from three independent groups.

## Rope climbing

Animals were trained to climb on a vertically strained customary cord (1.5 mm in diameter, length 100 cm) to reach a platform where they were rewarded with water and oatmeal (*Kashiwabuchi et al., 1995*). The time was recorded from first contact with the rope until the animals had reached the platform with all four limbs. Experiments were stopped after 60 s if the animal was not able to reach the platform within this time and the measure of 60 s was used for analysis. Each experiment consisted of five trials and the median value of the best three runs was used for further analysis. To assess climbing performance and coordination of the hind limbs, we developed a non-linear score ranging from 1 to 4 (*Video 1*): (1) The mouse uses all four limbs in an alternating

manner over the complete distance. (2) The mouse uses the hind limbs in an alternating manner more than half of the distance. (3) The mouse never uses the hind limbs in an alternating manner. (4) The mouse is not able to grasp the rope with the hind limbs. The median of the trials was used for further analysis. Videos were taken for each animal to confirm the scoring by another independent investigator who was blinded as to the genotype. Data for the 3-month time point were obtained from a single experimental group, and data for the 7- and 14-month time points were from three independent groups.

## Forelimb grip strength

Forelimb grip strength of $Cln3^{\Delta ex1-6}$ mice and wt littermates was tested using a grip strength meter (TSE Systems Inc.). Mice were held at the proximal part of the tail and allowed to grasp a horizontal metal bar. Mice were then pulled away and the pull force until the mice unclasp the bar was recorded. Eight trails within 2 min were performed for each mouse and median values were taken for further analysis. Data for the 3-month time point were obtained from a single experimental group, and data for the 7- and 14-month time points were from three independent groups.

## Gait analysis

Gait analysis was performed according to Kunkel-Bagden (**Kunkel-Bagden et al., 1993**) at the age of 7 and 14 months. In short, hind feet of mice were painted with blue ink and mice were then placed in a narrow tunnel so that the mice move straight forward and leave footprints on subjacent sheets of white paper. The parameters of the footprints (right and left stride length, stride width and angel between hind feet) were analyzed by a blinded and experienced experimenter.

## Open field

The OF was a square-shaped box (50 × 50 × 40 cm) made of Perspex XT, a black opaque material semi-permeable for infrared light (TSE Systems Inc). The apparatus was illuminated by infrared LEDs from below, and activity monitoring was conducted using an IR sensitive CCD camera (GKB Security Corporation, Taichung, Taiwan) along with the computerized video tracking software VideoMot2 (TSE Systems Inc.). The OF arena was subdivided into a center zone (25 × 25 cm) and the surrounding periphery. The illumination at the floor level slightly decreased from the center (200 lux) to the periphery (150 lux), due to shadows arising from the walls of the OF. Mice were placed into the periphery zone adjacent to the wall and allowed to freely explore the apparatus for 5 min. Parameters measured by the automated system included total distance travelled, time spent in either center or periphery zone, and number of entries into each zone. Additional parameters including rearing behavior and grooming activity were scored as indicators of explorative and anxiety-related behavior. After completion of the test, the OF was cleaned with 70% alcohol between trials ensuring identical experimental conditions for all subsequent tests. Data were obtained from two independent experimental groups.

## Elevated plus maze

The plus-shaped maze was made of grey Perspex (TSE Systems Inc.) and had two sets of opposing arms (30 × 5 cm) extending from a central platform (5 × 5 cm). Two arms were enclosed on either side by 15 cm high, opaque walls. The remaining two arms were open and surrounded by a slightly raised lip (0.25 cm). The device was elevated 50 cm above floor level. Illumination intensity was 200 lux on the open arms, 120 lux on the intersection, and 40 lux on the closed arms. Mice were placed in the central area, facing one of the open arms, and then allowed to freely explore the maze for 5 min. The number of arm entries, the amount of time spent in the open and closed arms, and the total distance travelled were recorded using an automated video tracking system (VideoMot2; TSE Systems Inc.). An arm entry was defined by all four feet being set in the respective arm. In addition, other behavioral parameters were scored manually, including rearing and grooming behavior. Before the next trial the EPM was thoroughly cleaned ensuring identical experimental conditions for all subsequent tests. Data were obtained from two independent experimental groups.

## Dish test

Mice were placed in a commercial Petri glass dish (150 × 15 mm) and the time was measured until the mice stepped out of the dish with all four feet. This latency is taken as an indicator for explorative behavior. Dishes were cleaned with 70% Ethanol between each trial. Data was obtained from two independent experimental groups.

## Fear conditioning

FC was performed in two experiments with two separate groups of animals. First experiment was performed using an automated fear conditioning system from TSE Systems Inc., and the second group was tested on a system from Ugo Basile S.R.L. (Gemonio, Italy) with the same testing parameters in both experiments. Since experience of aversive stimuli would compromise the results of repetitive testing FC was only performed once for each animal at the age of 14 months. All tests were video-recorded, and the videos from both experiments were automatically scored for freezing using the ANY-Maze software (Stoelting Co. Wood Dale, Illinois, USA) ; freezing detection threshold: 2 s) by the same blinded investigator. The testing chamber in both setups was illuminated with a light intensity of 100 lux throughout the entire experiment. During the conditioning phase (day 0), mice were placed in the conditioning chamber and allowed to briefly explore the novel environment for 2 min before a 80 dB strong 10 kHz sinusoidal tone was presented for 20 s as an auditory cue. During the last 2 s of the tone, a 0.75 mA foot shock was administered. An 80 s inter-trial interval preceded another identical trial. Following the last tone plus shock paired stimulus, mice remained in the chamber for 60 s before they were removed and placed back to the home cage. Fecal boli and urine were removed and the apparatus was cleaned with 70% alcohol.

Cued fear testing was performed 1 day and 7 days after the conditioning trial. Mice were tested in the same chamber, but in a modified, 'novel' environment. To achieve this, the grid floor was covered with a grey, opaque PVC plate with a handful of woodchip bedding. The Perspex arena was made opaque by covering it with black-and-white patterned stripes. A novel olfactory cue was provided using 3% acetic acid solution for cleaning instead of 70% alcohol. Mice were then gently placed into the chamber and allowed to explore the altered environment for 2 min. Subsequently, the auditory cue (80 dB, 10 kHz sinusoidal tone) was presented for 3 min and behavioral responses were recorded. After termination of the tone, mice remained in the chamber for 30 s before they were returned to the home cage.

Contextual fear testing was performed at 2 and 8 days after the conditioning session. Mice were placed into the original conditioning chamber in the absence of the tone for 5 min and behavioral responses were recorded. After removal of the animals the apparatus was cleaned.

During the entire behavioral experiments, responses of mice were monitored by an observer blinded as to genotype of the mice and videotaped in a coded way for double-check analysis. For each test, the time spent freezing and the latency to the first freezing event were scored manually following a standardized protocol. Data were obtained from two independent experimental groups.

## Neurophysiological recordings

### Whole-cell recordings of lateral amygdala projection neurons

Coronal slices (300 μm) containing the amygdala from 14-month-old mice were prepared on a vibratome, incubated at 34°C for 20 min, and stored thereafter at room temperature (RT). Single slices were placed at RT in a submersion chamber and were perfused with extracellular solution (aCSF-1) containing (in mM): 120 NaCl, 2.5 KCl, 1.25 NaH2PO4, 2 MgSO4, 2 CaCl2, 22 NaHCO3 and 20 glucose. The pH was set to 7.35 by continuously gassing with 95% $O_2$/5%$CO_2$. Whole-cell patch clamp recordings were performed using an EPC-10 patch-clamp amplifier at a sampling rate of 10 kHz. The recordings were done on principal neurons (PNs) in the lateral amygdala (LA), which were identified morphologically and by their action potential pattern in response to depolarizing current injections during current-clamp recordings. The PNs show a low number of generated action potentials and a typical adaptation of action potential generation as described previously (*Sah and Lopez De Armentia, 2003*; *Lange et al., 2012*). Patch-pipettes (2.2–2,8 MΩ; borosilicate glass; Clark Electromedical Instruments) for miniature (m), spontaneous (s) and evoked (e) inhibitory postsynaptic currents (IPSCs) were filled with a solution for each individual experiment as follows (mM): NaCl 10, KCl 110, EDTA 11, HEPES 10, MgCl2 1, CaCl2 0,5, Phosphocreatine 15, MgATP 3, and NaGTP 0.5. The pH

was set to 7.25. The isolated IPSCs were recorded in voltage-clamp mode at a membrane potential of −70 mV in the presence of AP5 (D-(-)−2-amino-5-phosphonopentanoic acid; 50 µM; Abcam) and DNQX (6,7-dinitroquinoxaline-2,3-dione disodium salt; 10 µM; Abcam) to block NMDA- and AMPA-receptors. For recording of miniature (m), spontaneous (s), and evoked (e) excitatory postsynaptic currents (EPSCs), patch-pipettes were filled with (mM): NaCl 10, K-gluconate 88, $K_3$-citrate 20, HEPES 10, BAPTA 3, Phosphocreatine 15, MgCl2 1, CaCl2 0,5, MgATP 3, and NaGTP 0.5. For isolation of glutamatergic EPSCs, recordings were performed with $GABA_B$-and $GABA_A$-receptors being blocked in the presence of CGP 52432 (3-[[(3,4-Dichlorophenyl)-methyl]amino]propyl(diethoxy-methyl)-phosphinic acid; 2 µM, Abcam) and gabazine (6-Imino-3-(4-methoxyphenyl)−1(6 hr)-pyridazi-nebutanoic acid hydrobromide; 25 µM; Abcam). EPSCs and IPSCs were recorded at a holding potential of −70 mV. Extracellular stimulation for the recording of evoked postsynaptic currents (ePSCs) was performed using a bipolar tungsten stimulation electrode, placed locally in the LA. Amplitudes of ePSCs were measured and averaged from five consecutive PSCs with an interval of 15 s. Values were averaged from different neurons and are presented as the mean amplitude (in pA). Miniature and spontaneous postsynaptic currents were recorded over a time period of 300 s (60 traces with 5 s duration). For measuring of mIPSCs and mEPSCs TTX (1 µM, Biotrend) was applied.

## Whole-cell recordings of dentate gyrus granule cells

Transversal hippocampal slices (300 µm) from 14-month-old mice were prepared on a vibratome, incubated in aCSF-2 (mM): 125 NaCl, 25 $NaHCO_3$, 25 Glucose, 2.5 KCl, 1.25 $NaH_2PO4$, 2 $CaCl_2$, 1 $MgCl_2$ at 32°C for 30 min, and subsequently stored at RT and continuously purged with 95% $O_2$/5% $CO_2$. For recordings slices were transferred to a recording chamber and continuously perfused with aCSF-2 at RT. Recordings were were made from visually identified dentate granule cells (GC) and were performed as described previously (Geis et al., 2010). In short, patch-clamp pipettes were pulled from borosilicate glass (3–3,7 MΩ; GB200F-10; Science Products, Hofheim, Germany) and filled with intracellular solution (mM): 145 KCl, 10 Hepes, 0,1 EGTA, 2 $Na_2ATP$, 2 $MgCl_2$ for the recording of IPSCs. Recordings of EPSCs were performed in aCSF-1 and the patch pipettes where filled with (mM): 115 K-gluconate, 40 KCl, 10 Hepes, 0.1 EGTA, 2 $Na_2ATP$, $MgCl_2$. IPSCs were recorded in the presence of CNQX (10 µM; 6-cyano-7-nitroquinoxaline-2,3-dione) and AP5 (50 µm mM; both Tocris Bioscience). For the recording of EPSCs, CGP 52432 and gabazine was added. For measuring of mIPSCs and mEPSCs TTX (1 µM) was applied. All recordings were performed at RT. For stimulation of afferent fibers borosilicate glass pipettes (4–5 MΩ) filled with aCSF-2 were used and placed in the granule cell layer (GCL) or the outer molecular layer (OML) to stimulate different types of inhibitory synaptic inputs onto the GC. For the recoding of eEPSC, the lateral perforant path was stimulated. The stimulation frequency was 0.3 Hz and stimulation strength was adjusted to the minimal strength for eIPSC. For analysis of eESPC after lateral perforant path stimulation, an in-out curve was conducted. For recording of paired pulse depression (PPD) of eIPSC stimulation strength was increased by 50% to reduce the number of stimulation failures. For analysis of paired pulse fascilitation of ESPC in lateral perforant path, the stimulation intensity was set to 100 pA.

Whole cell path clamp recordings in the amygdala and the hippocampus were excluded when the initial membrane potential was more positive than −60 mV or changed by more than 20% during recordings and when series resistance was more than 20 MΩ or changed by more than 20%.

## Loose patch recordings in cerebellar slices ex-vivo

Coronal slices (300 µm) of the dissected mouse cerebellum at the age of 14 months were prepared using a vibratome and were incubated in a preparation solution containing (in mM) 125 NaCl, 26 $NaHCO_3$, 20 Glucose, 2.5 KCl, 1.25 $NaH_2PO4$, 2 $CaCl_2$, 1 $MgCl_2$. After an incubation period of 30 min at 32°C, slices were transferred to a recording chamber with continuous perfusion of aCSF-2 at a constant flow rate of ~ 2 ml/min. All solutions were continuously purged with 95% $O_2$/5%$CO_2$. PCs were visually identified (assisted by DODT gradient optics, Luigs and Neumann, Ratingen, Germany) and recordings were performed with an EPC-10 amplifier (HEKA, Ludwigshafen, Germany). For recording spontaneous activity, loose patch recordings were made from PCs with low-resistance (2–2.5 MΩ) borosilicate pipettes filled with aCSF-2. Evoked responses were elicited by stimulating parallel fibers about 200 µm away from the PC bodies with an additional pipette containing aCSF-2. Stimuli (6–10V, 100µs) were applied via a triggered stimulus isolator (Isoflex, A.M.P.I.) at 0.5 Hz.

Data were filtered at 2.9 kHz and digitized at 10 kHz. Simple spike frequencies, inter-spike intervals (ISI), coefficients of variation of intervals (CV) and coefficients of adjacent intervals (CV$^2$; mean value of $2 \times \left[\frac{|ISI_{n+1} - ISI_n|}{(ISI_{n+1} + ISI_n)}\right]$) were analyzed using IgorPro 4.0 and 'Neuromatic' extension spike detection routine (WaveMetrics).

## Hofmann (H)-Reflex recording in-vivo

The H-reflex recordings were performed as previously described (*Geis et al., 2010*). Animals were deeply anesthetized and recordings of orthodromic distal motor responses and of H-reflex responses were made from the small foot muscles with steel needle electrodes using a Toennies electromyograph and NeuroScreenplus software (Erich Jaeger GmbH, Friedberg, Germany). Threshold currents for the first occurrence of a small M-response were determined usually resulting in 1–2 mA at 0.05 ms stimulus duration. Thereafter, the current was increased at 0.1 mA per stimulation until reproducible H-waves following the M-response were seen usually followed by F-waves (*Krieger et al., 2014*). Post-activation depression (*Lee et al., 2005*) was calculated for each frequency using the ratio of the H-reflex and the orthodromic motor response (H/M Ratio) of the 1st and the 10th response. Recordings with abnormally low amplitudes or non-reproducible recordings were discarded, as were animals showing signs of bleeding at the stimulation or recording sites.

## Dorsal root potential (DRP) recordings in vivo

In vivo recordings of DRP in mice were performed as described (*Grünewald and Geis, 2014*). Briefly, animals were deeply anesthetized and a dorsal laminectomy was performed from lumbar to thoracic levels to expose the spinal cord and dorsal roots and the dura mater was carefully removed. The exposed spinal cord was continuously covered by artificial cerebrospinal fluid (aCSF-3) (in mM: 134 NaCl, 3 KCl, 1.25 KH$_2$PO$_4$, 2 MgSO$_4$H$_2$O, 2.5 NaHCO$_3$, 2 CaCl$_2$, 10 glucose, 0.003% H$_2$O$_2$). Stimulation and recording of cut roots was performed using glass suction electrodes. Stimulation was performed with a Grass S88 stimulator (Grass Technologies Warwick, RI, USA; stimulus duration 0.2 ms, single stimulation and trains of three stimuli at 100 Hz) and potentials were recorded with an ELC-03X amplifier (NPI Electronic, Tamm, Germany) and analyzed with Patchmaster software (both HEKA). Stimulation strength was increased further by 30% above the current resulting in maximum amplitude (supramaximal stimulation). Recordings with movement artifacts or twitching of adjacent tissue were discarded. After these experiments, mice were sacrificed by decapitation in deep anesthesia and tissues were harvested.

## Histology

Thirty μm thick slices of frozen brain, cerebellar, and spinal cord tissue were made on a sliding microtome (HM 430 + fast freezing unit KS 34, Thermo scientific, Waltham, Massachusetts) and subsequently stored at −20°C in antifreeze solution. Staining was performed according to a standard protocol. Sections were incubated with primary antibodies over night at 4°C and with secondary antibodies for 2 hr at RT. The following antibodies and dilutions were used for all stains presented: anti-calbindin (Swant, Marly, Switzerland, 1:5000), anti-GAD65/67 (Merck, Darmstadt, Germany; 1:8000), anti-Neuropeptide Y (Thermo Fisher Scientific; 1:1000), anti-Calretinin (Swant; 1:5000), anti-Somatostatin (Acris Antibodies, Herfordt, Germany; 1:50), anti-Parvalbumin (Swant; 1:5000).

## Cell quantification

All cell counts were performed by investigators blinded to the genotype of the mice. The quantification of interneuron numbers in the hippocampus and the cerebellum was performed using an unbiased stereological method (optical fractionator, StereoInvestigator; Micro Brightfield, Williston, VT). 6–7 slices were analyzed per mouse (hippocamous: every eighth slice between bregma −1.34 and 2.8 mm for each staining; cerebellum: every eighth slice of coronar sections of the anterior vermis cortex). Region of interest was defined under a 5x objective. Stereological counting was performed with a 400x magnification for NPY and NeuN and 200x for all other stainings. Grid and counting frame size were adjusted for every staining to obtain at least 1 to 3 counts per dissector but kept constant for all animals per staining. The volume was calculated from areas of interest in each slice, the slice thickness, the number of slices and the section interval. The estimated population is given as cells*mm$^{-3}$.

For analysis of interneuron counts in the amygdala, stereology was not performed because the estimated coefficients of error where too high. This is due to the uneven distribution of cell types throughout the area of interest (contour of the amygdala) and its smaller size. To obtain reliable measurements for group comparison, interneurons were counted in unbiased fashion after the following procedure. Cell counting was performed by the same investigator blinded to the experimental groups and to the genotype of each animal. We performed quantification in 5–7 transversal sections spanning the entire amygdala complex from bregma −0.8 mm to −2.3 mm assuring that not only subfields of the amygdala are analyzed. Each section for every interneuron and NeuN stain was separated by seven adjacent sections (spacing of 210 µm) so that oversampling of neurons, also of probably enlarged interneurons is avoided. We used an Axioplan two microscope, 20x Plan Neofluar objective, AxioCam MRc 5, Axovision software, and MosaiX module (Zeiss, Oberkochen, Germany) and for each of the 5–7 slices per staining compound images were made. Contours of the area of interest were drawn and cells were counted offline using ImageJ. Nuclei intersecting the uppermost focal plane and the lateral boarder areas were excluded from the count to avoid oversampling. The density of immunoreactive neurons for each animal was determined by dividing the total number of positive cells by the total sampled surface. Cell density (Cells per mm$^2$) was calculated separately for each slice and median value for each mouse brain was used for statistical analysis.

## Statistical analysis

Statistical analyses were performed using SPSS software (IBM) and Sigmaplot 12 (Systat Software Inc.). Sample size calculation was performed using Sigmaplot 12 sample size function with an alpha-value of 0.05 and a desired power (beta value) of 0.8. Difference in mean and standard deviation estimates were taken from previous experiments using the respective methods. Parametric or non-parametric statistical tests were used as indicated according to the distribution of the data which was evaluated with the Shapiro-Wilk test. All dispersion and precision measures, as well as exact p values are listed in *Supplementary file 2*. Cohen's d as a measure of effect size was calculated as $d = \frac{\bar{x}_1 - \bar{x}_2}{s}$ with $s = \sqrt{\frac{(n_1-1)s_1^2 + (n_2-1)s_2^2}{n_1+n_2-2}}$ and $s_n^2 = \frac{1}{n_1-1}\sum_{i=1}^{n_1}\left(x_{n,i} - \bar{x}_n\right)^2$. Asterisks indicate the significance level (*p<0,05, **p<0,01, ***p<0,001). Data are reported as means ± SEM.

## Acknowledgements

We thank B Dekant, S Hellmig, H Brünner (Würzburg), and C Sommer (Jena) for providing expert technical assistance in animal experiments and immunohistochemistry. We thank R Martini and J Groh (Würzburg) for providing aged *Cln3*$^{\Delta ex1-6}$ mice. The authors declare no competing financial interest. This work was supported by the Deutsche Forschungsgemeinschaft (SFB 581 [TP A7], SFB/TR 166 [B2], SFB/TR58 [A03], GE2519_3–1), by the IZKF and CSCC Jena, and by intramural University Research Funds (Würzburg and Jena). KVT is a Senior Research Professor funded by intramural funds from the Würzburg University Medical School.

## Additional information

### Funding

| Funder | Grant reference number | Author |
| --- | --- | --- |
| Deutsche Forschungsgemeinschaft | SFB 581 [TP A7] | Klaus V Toyka<br>Claudia Sommer |
| Deutsche Forschungsgemeinschaft | SFB/TR 166 [B2] | Christian Geis |
| Deutsche Forschungsgemeinschaft | SFB/TR 58 | Maren D Lange<br>Hans C Pape |
| Bundesministerium für Bildung und Forschung | Center for Sepsis Control and Care | Christian Geis |
| IZKF University Hospital Jena | | Benedikt Grünewald<br>Christian Geis |

The funders had no role in study design, data collection and interpretation, or the decision to submit the work for publication.

### Author contributions

Benedikt Grünewald, Conceptualization, Data curation, Formal analysis, Investigation, Methodology, Writing—original draft, Project administration, Writing—review and editing; Maren D Lange, Data curation, Investigation, Visualization, Writing—original draft, Writing—review and editing; Christian Werner, Investigation, Visualization, Methodology, Writing—original draft; Aet O'Leary, Investigation, Methodology, Writing—original draft; Andreas Weishaupt, Resources, Methodology; Sandy Popp, Investigation, Analysis and interpretation of data, Revising the article critically for important intellectual content, Final approval of the version to be published; David A Pearce, Resources, Substantial contributions to conception and design, Revising the article critically for important intellectual content, Final approval of the version to be published; Heinz Wiendl, Resources, Analysis and interpretation of data, Revising the article critically for important intellectual content, Final approval of the version to be published; Andreas Reif, Resources, Project administration; Hans C Pape, Conceptualization, Resources, Writing—review and editing; Klaus V Toyka, Claudia Sommer, Conceptualization, Resources, Supervision, Writing—review and editing; Christian Geis, Conceptualization, Supervision, Investigation, Writing—original draft, Writing—review and editing

### Author ORCIDs

Aet O'Leary (iD) https://orcid.org/0000-0001-6783-4729
Christian Geis (iD) http://orcid.org/0000-0002-9859-581X

### Ethics

Animal experimentation: All animal experiments were approved by the respective Bavarian and Thuringian state authorities (No. 55.5-2531.01-12/10; 78/05 and 02-44/12). All efforts were made to minimize animal suffering and to reduce the number of animals used. The study was performed in accordance with the ARRIVE guidelines for reporting animal research (Kilkenny et al., 2010).

### Decision letter and Author response

Decision letter https://doi.org/10.7554/eLife.28685.026
Author response https://doi.org/10.7554/eLife.28685.027

## Additional files

### Supplementary files

• Supplementary file 1. Animal groups and counts. List of all animal groups used in the study including animal number and the experiments in which they were included. Note that not all animals in the list were finally included in the analysis (e.g. mice were allocated to different tests, exclusion due to unstable recordings or insufficient quality criteria for neurophysiology etc.). The exact number of mice used in the respective experiment is given in the figure legends. $Cln3^{\Delta ex1-6}$ mice and age-matched littermates where allocated in a randomized way to the respective groups before starting the experiments by a blinded member of the laboratory.
DOI: https://doi.org/10.7554/eLife.28685.022

• Supplementary file 2. Statistic report table. Table containing information regarding distribution, statistical analysis method, exact p values, n-values and number of biological replicates, mean, SD, SEM, median and confidence intervals, and Cohen's d effect size for all experiments.
DOI: https://doi.org/10.7554/eLife.28685.023

• Transparent reporting form
DOI: https://doi.org/10.7554/eLife.28685.024

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
