## [Decision Letter]

Thank you for submitting your article "Defective synaptic transmission causes disease signs in a mouse model of Juvenile Neuronal Ceroid Lipofuscinosis" for consideration by *eLife*. Your article has been reviewed by two peer reviewers, and the evaluation has been overseen by a Reviewing Editor and a Senior Editor. The following individual involved in review of your submission has agreed to reveal their identity: Jonathan D. Cooper (Reviewer #2).

The reviewers have discussed the reviews with one another and the Reviewing Editor has drafted this decision to help you prepare a revised submission.

Summary:

This manuscript by Grünewald et al. describes a comprehensive series of carefully conducted experiments revealing evidence for increased anxiety and impaired synaptic transmission in a mouse model of juvenile Batten disease (JNCL), a fatal inherited neurodegenerative storage disorder that affects children and young adults. The experimental work comprises a remarkable amount of data, ranging from state-of-the-art electrophysiology to animal behavior to histology. Using a *cln3* knockout mouse model they provide detailed neurophysiological analyses in together 4 different functional systems in the CNS (hippocampus, amygdala, cerebellum, spinal cord). They uncover disturbed GABAergic and glutamatergic transmission in 3 of these areas (excluding spinal cord) as causative pathomechanisms underlying characteristic disease symptoms. Functional experiments are paralleled by extensive behavioral analyses and quantitative immunohistology showing loss of certain subtypes of interneurons. Together, this is a complete work on many functional and cellular aspects in the *cln3* mouse model. The findings are important for understanding this devastating disorder of children and young adults and set a milestone in research on JNCL. The reviewers had rather minor issues that need to be addressed by the authors.

Essential revisions:a) The data clearly reveal increased anxiety by both dish test and more conventional open field and elevated plus maze testing. Although the age at which dish testing was conducted is clearly stated, it is not immediately apparent what age the open field and elevated plus maze testing were performed at (without referring to the Materials and methods). It would also be informative to know if these phenotypes remained stable with time or if they worsened with disease progression.

b) The electrophysiological characterization of pathways in the lateral amygdala reveal clear evidence for impaired synaptic transmission. These broadly point to defects in GABAergic with both miniature and spontaneous post synaptic currents both reduced in frequency, and in the case of sIPSCs in amplitude too. There is also clear evidence for an impact of *Cln3* deficiency upon postsynaptic excitatory potentials, but once again it is not immediately apparent what age of mice the recordings were taken from or if the magnitude of these defects in synaptic transmission worsen with disease progression. This is particularly important for the interpretation of the pathology data presented in Figure 3, which is from 13 month old mice.

c) The effects on interneuron survival are clearly documented, the authors mention that they did not perform stereological analysis due to the uneven distribution of these cells and their relative paucity. If a structure is sampled adequately then this should overcome any uneven distribution of cells, but it is acknowledged that very few cells being present renders stereological counts unfeasible. Nevertheless, the authors mention that the counts were made in an 'unbiased fashion', but it is not clear exactly how this is done. Please could they clarify this? One confounding issue would be the potential effects on interneuron size (likely to increase due to accumulated storage material), which may bias any non-stereological counts (unless they are corrected for oversampling using any one of the traditional methods for doing this). It would also be good to have data for the overall number of neurons (or at least the number of excitatory neurons) so that the selectivity of these effects could be judged.

d) Given the clear data for altered synaptic transmission it would be informative, but not essential, to determine if staining for both pre- and post-synaptic markers were changed in the amygdala and hippocampus. Knowing the time course of such synaptic pathology would be important for interpreting the detailed electrophysiological data presented.

e) While the contextual fear and electrophysiological data from the hippocampus are novel and informative, it would again be helpful to more clearly state what age of animal these studies were performed in and whether these might worsen with time. This information would be important for interpreting the relationship between these effects upon behavior and synaptic transmission to the cellular pathology documented in the hippocampus. In relation to these data, although there appear to be selective effects on both NPY and parvalbumin inhibitory interneurons (similar to those reported previously), it is unclear whether these effects are specific to any particular subfield, or whether the survival of excitatory neurons in these subfields is similarly affected.

f) Since rotarod testing likely involves a variety of behavioral responses, it is good to see the use of another apparently more reliable test of motor performance in the form of a climbing up a taught cord. This mirrors vertical pole testing, which can also reveal defects in motor co-ordination in these mice. Did the authors consider (or test) whether similar defects in the ability to descend a cord were apparent in *Cln3* mice? Rotarod testing is notoriously variable in terms of the effects seen in different labs, and the data presented here are markedly different from those previously published by one of the authors in other studies. Can the authors comment on this disparity? The results of walking distance are very interesting, but did the authors observe any gait abnormalities in these mice?

g) The data for altered firing of cerebellar Purkinje cells are both robust and highly interesting in likely revealing a possible explanation for the impaired co-ordination of *Cln3* mice (especially in the absence of any overt effects on spinal inhibition – very interesting in itself given a recent paper documenting spinal pathology in another form of NCL). The authors should also assess interneuron survival in the cerebellum (as they have in the amygdala and hippocampus) to investigate the cellular substrate for these electrophysiological phenotypes.

h) Why did the authors choose to use the *Cln3* null mutant mouse (more accurately *Cln3^∆ex1-6^* rather than *Cln3^-/-^*) for these studies, when the majority of investigators have switched to using the *Cln3^∆ex7/8^* mouse that recapitulates the most common disease-causing mutation in JNCL? While the phenotypes of this mice do not appear to differ much (if at all), it would be good to state this somewhere in the manuscript.

i) Amount and diversity of electrophysiological measurements is impressive; could the authors provide exemplary information on resting membrane potential and input resistance of patch-clamp recordings to get an impression if the basic properties of neurons are different in wildtype and knockout animals?

---

## [Author Response]

Essential revisions:a) The data clearly reveal increased anxiety by both dish test and more conventional open field and elevated plus maze testing. Although the age at which dish testing was conducted is clearly stated, it is not immediately apparent what age the open field and elevated plus maze testing were performed at (without referring to the Materials and methods). It would also be informative to know if these phenotypes remained stable with time or if they worsened with disease progression.

We have now clearly stated the age of the mice for open field and elevated plus maze testing (14 months) in Figure 1 and in the figure legend. Additionally, we have now included data of open field and elevated plus maze testing at 7 months (see new Figure 1—figure supplement 1). In younger mice there is no difference between *cln3* knockout (*Cln3^∆ex1-6^*) and wt littermates. Obviously, anxiety-related behavior worsened with disease progression, similar as compared to the coordinative dysfunction of *cln3* knockout mice. This is now also mentioned in the Discussion (subsection “Reduced learning abilities and increased anxiety behavior in *Cln3^Δex1-6^* mice is based on defective synaptic transmission”)

b) The electrophysiological characterization of pathways in the lateral amygdala reveal clear evidence for impaired synaptic transmission. These broadly point to defects in GABAergic with both miniature and spontaneous post synaptic currents both reduced in frequency, and in the case of sIPSCs in amplitude too. There is also clear evidence for an impact of Cln3 deficiency upon postsynaptic excitatory potentials, but once again it is not immediately apparent what age of mice the recordings were taken from or if the magnitude of these defects in synaptic transmission worsen with disease progression. This is particularly important for the interpretation of the pathology data presented in Figure 3, which is from 13 month old mice.

We apologize for not having mentioned the age of amygdala recording in the main text; in principle, all electrophysiological recordings (amygdala, hippocampus, cerebellum, and spinal cord) were performed at the age of 14 months (unless not otherwise stated). Similarly, all histopathological analyses were performed at the age of 14 months, so that electrophysiological and histopathological data are obtained from mice of the same age. This is now clearly indicated in the figure legends, Results section, and in the Materials and methods section.

There is clear evidence for progressive dysfunction obtained from extensive behavioral testing. Histological and electrophysiological data are from 14 months old mice where the behavioral phenotype is most pronounced.

c) The effects on interneuron survival are clearly documented, the authors mention that they did not perform stereological analysis due to the uneven distribution of these cells and their relative paucity. If a structure is sampled adequately then this should overcome any uneven distribution of cells, but it is acknowledged that very few cells being present renders stereological counts unfeasible. Nevertheless, the authors mention that the counts were made in an 'unbiased fashion', but it is not clear exactly how this is done. Please could they clarify this? One confounding issue would be the potential effects on interneuron size (likely to increase due to accumulated storage material), which may bias any non-stereological counts (unless they are corrected for oversampling using any one of the traditional methods for doing this). It would also be good to have data for the overall number of neurons (or at least the number of excitatory neurons) so that the selectivity of these effects could be judged.

Quantification in the amygdala was performed in an unbiased fashion taking into account all the considerations of the reviewer. First, cell counting was performed by the same investigator blinded to the experimental groups and to the genotype of each animal. Second, we performed quantification in 5-7 transversal sections spanning the entire amygdala complex assuring that not only subfields of the amygdala are analyzed. Here, each section for every interneuron stain was separated by 7 adjacent sections (spacing of 210 µm) so that oversampling of neurons, even of probably enlarged interneurons is avoided. Third, nuclei intersecting the uppermost focal plane and the lateral boarder areas were excluded from the count to avoid oversampling. The density of immunoreactive neurons for each animal was determined by dividing the total number of positive cells by the total sampled surface. Cell density (Cells per mm2) was calculated separately for each slice and median value for each mouse brain was used for statistical analysis. This is now stated in detail in the Materials and methods section (subsection “Cell quantification”).

We performed complete new series of NeuN stains and quantitative analysis in the amygdala, hippocampus (subfields CA1, CA3, dentate gyrus) and cerebellum to obtain quantitative data of the overall number of neurons in these areas. Here, we did not find any differences in cell count in all of these areas, thus suggesting a primary effect on the respective interneuron subtypes. This new information is shown in these new figures: Figure 8; Figure 3—figure supplement 1; Figure 5—figure supplement 2 and discussed in the revised version of the manuscript (subsection “Loss of interneurons in the amygdala and hippocampus directly contributes to defective GABAergic signaling and affects local networks *Cln3^Δex1-6^* mice”, first paragraph).

d) Given the clear data for altered synaptic transmission it would be informative, but not essential, to determine if staining for both pre- and post-synaptic markers were changed in the amygdala and hippocampus. Knowing the time course of such synaptic pathology would be important for interpreting the detailed electrophysiological data presented.

Indeed, we found clear alteration of synaptic transmission as we showed in electrophysiological recordings in the amygdala, hippocampus, and cerebellum. We discussed a possible role of *CLN3* protein in neuronal transmission (see Discussion subsection “Reduced learning abilities and increased anxiety behavior in *Cln3^Δex1-6^*mice is based on defective synaptic transmission”, last paragraph). We agree with the reviewer that evaluation of pre- and postsynaptic markers over time would be an interesting topic. However, we think that careful evaluation of synaptic histopathology is clearly a topic for following studies. In the current study we focus on the functional changes that we addressed by extensive electrophysiology and on detailed, quantitative analysis of neuronal loss in these areas.

e) While the contextual fear and electrophysiological data from the hippocampus are novel and informative, it would again be helpful to more clearly state what age of animal these studies were performed in and whether these might worsen with time. This information would be important for interpreting the relationship between these effects upon behavior and synaptic transmission to the cellular pathology documented in the hippocampus. In relation to these data, although there appear to be selective effects on both NPY and parvalbumin inhibitory interneurons (similar to those reported previously), it is unclear whether these effects are specific to any particular subfield, or whether the survival of excitatory neurons in these subfields is similarly affected.

Contextual fear analysis and electrophysiological recordings in the hippocampus were performed at the age of 14 months; please see also comment to question b). This is now clearly stated in the figure legend, Results, and Materials and methods section. Different from all other behavioral tests, we do not have sequential information of fear-related behavior at different ages. Since experience of aversive stimuli would compromise the results of repetitive testing at later time points, repetitive testing for fear conditioning is not possible. We have now included a short explanation in the Material and methods section (subsection “Fear Conditioning (FC)”, first paragraph).

We performed large series of new immunostains and assessed the number of NeuN positive neurons in the hippocampus, amygdala, and cerebellum. Stereological analysis showed unchanged NeuN positive cell count in all of these regions, also in the subfields(CA1, CA3, dentate gyrus) of the hippocampus (see new Figure 5—figure supplement 2). In addition to the overall cell counts in the hippocampus we now also show the data for interneuron subclasses in the dentate gyrus, as well as the CA1 and CA3 subfield of the hippocampus separately (see new Figure 5—figure supplement 1). Parvalbumin and NPY positive interneurons were primarily reduced in the CA3 subregion whereas GAD positive interneurons were significantly reduced in the CA1 region. However, analysis showed trends also in the other subregions of the hippocampus toward reduction of these interneurons subclasses as well. Therefore, we suggest not over-interpreting the reduction of interneuron subclasses to a certain subregion of the hippocampus. We inserted a short statement in the Results section (subsection “Hippocampal synaptic transmission is impaired in *Cln3^Δex1-6^* mice and cued and contextual fear memory is reduced”, last paragraph).

f) Since rotarod testing likely involves a variety of behavioral responses, it is good to see the use of another apparently more reliable test of motor performance in the form of a climbing up a taught cord. This mirrors vertical pole testing, which can also reveal defects in motor co-ordination in these mice. Did the authors consider (or test) whether similar defects in the ability to descend a cord were apparent in Cln3 mice? Rotarod testing is notoriously variable in terms of the effects seen in different labs, and the data presented here are markedly different from those previously published by one of the authors in other studies. Can the authors comment on this disparity? The results of walking distance are very interesting, but did the authors observe any gait abnormalities in these mice?

We agree with the reviewer that the rope climbing test or the equivalent vertical pole test reveals more reliable results as compared to the RotaRod testing. This is most likely due to the higher demand on motor function and coordinative movements. We performed the rope climbing test in a way that the mice have to climb up 100 cm of vertical strained rope. We suppose that similar dyscoordinative behavior would also be present when mice would climb down the rope, however, we did not formally test this.

The reviewer is right, RotaRod testing is quite variable depending on a variety of influential factors. As an example, in a previous study (Kovacs and Pearce, 2015), *Cln3^∆ex1-6^* and *Cln3^∆ex7/8^* mice were tested on two different genetic backgrounds, 129S6/SvEv and C57BL/6J, on the rotarod at 3 and 6 months of age. Only *Cln3^∆ex1-6^* males on the 129S6/SvEv background displayed a deficit in the rotarod test (at both 3 and 6 months). Furthermore, it is known that environmental factors have a strong influence on the behavioral test results. We have previously shown that environmental factors specific to a geographical location can change the body weight, motor coordination and motor learning capability of wild type mice commonly used as controls in transgenic studies (Kovacs and Pearce, 2013). In that study, we found that the type of mouse diet was partially responsible for the location-specific differences.

Moreover, we have now included additional data from gait analysis at the age of 7 and 14 months showing unchanged stride length, distance and angle between the hind limbs (new Figure 6—figure supplement 2, Results: subsection “Cerebellar dysfunction underlying motor deficits in *cln3^Δex1-6^* mice”, first paragraph, Materials and methods subsection “Gait analysis”).

g) The data for altered firing of cerebellar Purkinje cells are both robust and highly interesting in likely revealing a possible explanation for the impaired co-ordination of Cln3 mice (especially in the absence of any overt effects on spinal inhibition – very interesting in itself given a recent paper documenting spinal pathology in another form of NCL). The authors should also assess interneuron survival in the cerebellum (as they have in the amygdala and hippocampus) to investigate the cellular substrate for these electrophysiological phenotypes.

We performed additional series of immunohistological stains in the cerebellum and quantified interneurons using stereological analysis. Here, we analyzed the relevant subclasses of interneurons in the cerebellar cortex of the vermis (GAD65 and parvalbumin), the amount of Purkinje cells and the overall number of neurons (NeuN) using stereology.

We found reduction of GAD65 positive interneurons in the *Cln^3∆ex1-6^* mice whereas the number of Parvalbumin positive interneurons and Purkinje cells as well as the total number of neurons was unchanged. This is in line with the electrophysiological findings showing disturbed feedforward inhibition mediated by dysfunctional molecular layer interneurons modulating Purkinje cell spiking properties. These new analyses are now included in the manuscript (New Figure 8; Results: subsection “Cerebellar dysfunction underlying motor deficits in *cln3^Δex1-6^* mice”, last paragraph).

h) Why did the authors choose to use the Cln3 null mutant mouse (more accurately Cln3^∆ex1-6^ rather than Cln3^-/-^) for these studies, when the majority of investigators have switched to using the Cln3^∆ex7/8^ mouse that recapitulates the most common disease-causing mutation in JNCL? While the phenotypes of this mice do not appear to differ much (if at all), it would be good to state this somewhere in the manuscript.

The reviewer is right – to be more accurate, we replaced the term *cln3^-/^*^-^ by *Cln3^∆ex1-6^* throughout the manuscript and in the figures. We also agree with the reviewer that the behavioral differences in both mutants are similar and may only differ in the severity of behavioral abnormalities. In a previous study, when *Cln3^∆ex1-6^* and *Cln3^∆ex7/8^* mice were compared on two different genetic backgrounds, 129S6/SvEv and C57BL/6J, in the dish test, in a modified vertical pole test, rotarod test, and tail suspension test at 1, 3 and 6 months of age, *Cln3^∆ex1-6^* males on the 129S6/SvEv background showed the most severe phenotypes (Kovacs and Pearce, 2015). *Cln3^∆ex1-6^* males on the C57BL/6J background and *Cln3^∆ex7/8^* males on the 129S6/SvEv background also had well-defined behavioral deficits. *Cln3^∆ex7/8^* mice on the C57BL/6J background, which are used by other investigators, however, had only mild if any behavioral deficits (at least in our hands). We inserted a short statement in the Materials and methods part (subsection “Animals and experimental groups”, first paragraph).

i) Amount and diversity of electrophysiological measurements is impressive; could the authors provide exemplary information on resting membrane potential and input resistance of patch-clamp recordings to get an impression if the basic properties of neurons are different in wildtype and knockout animals?

We have evaluated resting membrane potential and input resistance of patch-clamp recordings in the amygdala and hippocampus. These basic properties of neurons were unchanged in the *Cln3^∆ex1-6^*mice and wt littermate controls. Exact values are provided in the Results section (subsection “Synaptic transmission defects in the amygdala are associated with anxiety related behavior in *Cln3^Δex1-6^*mice”, second paragraph and subsection “Hippocampal synaptic transmission is impaired in *Cln3^Δex1-6^*mice and cued and contextual fear memory is reduced”, first paragraph).